# Kernel Complexity Reduced Graph Contrastive Learning for Noisy Node Classification

## Abstract

Graph Neural Networks (GNNs) have achieved remarkable success in learning node representations and have demonstrated strong performance on node classification. However, their effectiveness can be substantially compromised by noise in real-world graph data. To address this challenge, we propose Kernel Complexity Reduced Graph Contrastive Learning (KCR-GCL), a principled framework for noisy node classification with a provable transductive generalization guarantee. KCR-GCL introduces a novel KCR-GCL encoder, which incorporates a new KCR self-attention layer that adaptively balances different frequency components of the graph inspired by generalized graph convolution and reduces the kernel complexity for provably improved generalization for transductive learning. The KCR-GCL encoder is optimized with a low-rank regularization term through the truncated nuclear norm (TNN) on the gram matrix of the learned features. The learned low-rank representations are then used to train a linear classifier for transductive node classification in noisy graph data. The design of KCR-GCL is inspired by the Low Frequency Property (LFP) widely studied in general deep learning and node-level graph learning, and is further supported by a sharp generalization bound for transductive learning. To the best of our knowledge, KCR-GCL is among the first to theoretically reveal the benefits of low-rank regularization in transductive settings for noisy graph data. Experiments on standard benchmarks highlight the effectiveness and robustness of KCR-GCL in learning node representations under noisy conditions. The code of KCR-GCL is available at https://anonymous.4open.science/status/KCR-GCL.

## 1 Introduction

Graph Neural Networks (GNNs) are widely recognized as effective tools for node representation learning (Kipf & Welling, 2017; Bruna et al., 2014; Hamilton et al., 2017; Xu et al., 2019b). However, the majority of existing GNN methods do not adequately address the presence of noise in the graph data (Zhu et al., 2024; Zhong et al., 2019). Such noise can arise either in the attributes or labels of nodes, introducing attribute noise and label noise, respectively. Prior studies (Patrini et al., 2017) have demonstrated that noise in the input data can significantly impair the generalization ability of neural networks. This issue is further amplified since noise associated with a few nodes can spread through the graph structure and affect other nodes (Dai et al., 2021; Wang et al., 2023; 2024c) in graph-structured data. As a result, corrupted nodes not only degrade their own representations but also influence those of their neighbors. Such a challenge highlights the need for GNN models that can learn effectively even in the presence of noisy inputs.

To this end, we introduce Kernel Complexity Reduced Graph Contrastive Learning, or KCR-GCL, which introduces a novel KCR-GCL encoder improving both robustness and generalization for node representation learning. The node representations learned by the KCR-GCL encoder are used by a linear transductive classifier for transductive node classification. Traditional strategies for robust learning either modify the loss function to accommodate corrupted data (Patrini et al., 2017; Goldberger & Ben-Reuven, 2017) or remove samples suspected to be noisy (Malach & Shalev-Shwartz, 2017; Jiang et al., 2018; Yu et al., 2019; Li et al., 2020; Han et al., 2018). Although such ideas have been adapted for graph settings (Dai et al., 2021; Qian et al., 2023; Zhuang & Al Hasan, 2022), they often depend on heuristics and lack theoretical backing in the transductive setting. In contrast, our

KCR-GCL encoder is empirically inspired by the low-frequency nature of graph signals, and theoretically supported by a new and sharp generalization bound developed for transductive learning. Our generalization bound features three components capturing the training loss of the classifier when using clean labels, the impact of label noise on the classification loss, and the kernel complexity of the gram matrix of the learned features. To the best of our knowledge, this paper is among the first to provide a principled theoretical justification for the advantage of low-rank representation learning with GCL and graph convolution under noisy graph data. Experimental evaluations conducted on widely used benchmarks demonstrate that KCR-GCL consistently achieves strong robustness and outperforms the current state-of-the-art. Although GNNs are known to function as low-pass filters, they do not explicitly target low-frequency signals. Consequently, their ability to exploit the Low-Frequency Property (LFP) widely studied in deep learning (Rahaman et al., 2019; Arora et al., 2019; Cao et al., 2021; Choraria et al., 2022; Wang et al., 2024b; 2025) in the presence of noisy graph data remains limited. As illustrated in Figure 1, deferred to Section 4.1, the LFP reveals that clean label information tends to concentrate within the low-rank part of the feature gram matrix. Unlike conventional GNNs, the KCR-GCL encoder explicitly captures LFP by learning low-rank node representations. Prior works such as (Cheng et al., 2021) have illustrated the benefit of such low-rank learning in mitigating attribute noise by introducing learnable filtering mechanisms. In comparison, the KCR-GCL encoder explicitly promotes low-frequency information through the low-rank regularization by the Truncated Nuclear Norm (TNN), which aligns with the LFP.

The node representations by our KCR-GCL encoder are generated through a novel KCR self-attention layer that explicitly balances low-frequency and high-frequency components of the graph. Inspired by polynomial graph filters commonly used in graph signal processing (Choi et al., 2024; Zhang et al., 2024a; Marques et al., 2020), the KCR self-attention layer learns to combine multiple powers of the attention weight matrix, enabling the model to adaptively emphasize structural patterns across different spectral ranges while improving the generalization capability of the KCR-GCL encoder. Recent studies involving graph attention and transformer-based architectures emphasize the need to balance both low- and high-frequency components for improved node representations (Choi et al., 2024; Zhang et al., 2024a). Although GFSA (Choi et al., 2024) and HONGAT (Zhang et al., 2024a) also learn to combine different powers of the attention matrix based on the generalized graph convolution (Marques et al., 2020), their learning objective is solely the cross-entropy loss on the training labels alone. In contrast, the KCR self-attention layer in our KCR-GCL encoder roots in our novel theoretical result about the generalization bound for transductive learning, and it offers a principled balance between low-frequency and high-frequency by reducing the principled and well-defined kernel complexity, aiming for the provable performance improvement for transductive learning through the TNN regularizer. Reduction of the TNN leads to the reduction of the kernel complexity, leading to lower generalization error bound for transductive learning thus better performance of node classification. Performance comparisons in Table 1 of Section 5.2 demonstrate that the KCR-GCL encoder outperforms the current state-of-the-art based on attention and transformers, GFSA (Choi et al., 2024) and HONGAT (Zhang et al., 2024a), when evaluated under label and attribute noise. As shown in Table 3 of Section 5.4, the KCR-GCL encoder achieves a lower kernel complexity, resulting in a lower upper bound for the test loss of transductive node classification, compared to competing graph contrastive and attention-based methods.

## 1.1 CONTRIBUTIONS

Our contributions are as follows.

First, we introduce Kernel Complexity Reduced Graph Contrastive Learning, or KCR-GCL, that learns robust node representations by a novel KCR-GCL encoder. The learned node representations are subsequently used by a linear classifier for transductive node classification. The KCR-GCL encoder features a novel KCR self-attention layer, which explicitly learns to balance low-frequency and high-frequency components in the graph, inspired by the generalized graph convolution (Marques et al., 2020). The optimization of the KCR-GCL encoder incorporates the TNN on the gram matrix of the learned features as a low-rank regularization term into the standard prototypical GCL objective. The design of the KCR-GCL encoder is motivated by the LFP, which shows that a low-rank projection of the clean label matrix captures most of its informative content. In contrast, label noise tends to spread uniformly across all eigenvectors of the classification kernel matrix.

Second, we provide a rigorous theoretical analysis that establishes generalization guarantee for the linear transductive classifier trained on the low-rank node representations produced by the KCR-

GCL encoder. In particular, we derive a novel and sharp upper bound on the test loss of unlabeled nodes. To the best of our knowledge, this is among the first results to theoretically demonstrate the advantage of learning low-rank node representations for robust transductive classification under noisy conditions. Moreover, our theoretical result establishes the connection between the generalization guarantee and the TNN regularizer, that is, reduced TNN indicates reduced kernel complexity and lower generalization error bound for transductive learning. Furthermore, the KCR self-attention balances different frequency components of the graph and sharpens the derived upper bound through even lower kernel complexity than that without KCR self-attention. As demonstrated in Table 3 of Section 5.4, the KCR-GCL encoder renders a lower kernel complexity and generalization upper bound than existing methods. Comprehensive experiments conducted on widely used graph benchmarks in Section 5.2 demonstrate the superiority of the KCR-GCL encoder over existing methods in node classification tasks involving noisy graph data.

## 2 RELATED WORKS

### 2.1 GRAPH NEURAL NETWORK AND ITS TRAINING ON NOISY DATA

The increasing adoption of contrastive learning has significantly advanced unsupervised representation learning on graphs (Suresh et al., 2021; Thakoor et al., 2021; Li et al., 2024a; Lee et al., 2022; Feng et al., 2022a; Zhang et al., 2023; Lin et al., 2023). In the graph domain, a large number of graph contrastive learning (GCL) methods (Velickovic et al., 2019; Sun et al., 2020; Hu et al., 2020b; Jiao et al., 2020; Peng et al., 2020; You et al., 2021; Jin et al., 2021; Mo et al., 2022) work by maximizing agreement between corresponding node embeddings across augmented views. Several recent works have incorporated semantic prototypes (Snell et al., 2017; Arik & Pfister, 2020; Allen et al., 2019; Xu et al., 2020) into the contrastive objective (Xu et al., 2021; Guo et al., 2022; Li et al., 2021). Meanwhile, GNNs remain central to node representation learning (Bruna et al., 2014; Kipf & Welling, 2017; Hamilton et al., 2017; Veličković et al., 2018; Xu et al., 2019b). However, it has been well established that GNNs are inherently vulnerable to noisy inputs, such as corrupted labels or features (Zhang et al., 2021). To improve robustness, prior work has explored loss correction, which modifies the training objective to account for label noise (Patrini et al., 2017; Goldberger & Ben-Reuven, 2017), and sample selection, which focuses training on selected clean samples (Malach & Shalev-Shwartz, 2017; Jiang et al., 2018; Yu et al., 2019; Li et al., 2020; Han et al., 2018). Within graph-based learning, robustness has also been addressed through label denoising, structural regularization, and auxiliary self-supervised tasks (Dai et al., 2021; Qian et al., 2023; Zhuang & Al Hasan, 2022; Li et al., 2024b; Yuan et al., 2023). While these methods build on external objectives or correction heuristics, our approach introduces a new perspective of enhancing GNN robustness by directly integrating low-rank regularization into the encoder training process of GCL.

### 2.2 BALANCING THE FREQUENCY COMPONENTS WITH GRAPH ATTENTION

The low-frequency bias of GNNs emphasizes the importance of leveraging smooth, low-frequency components embedded in both graph topology and node features (NT & Maehara, 2019; Xu et al., 2019a; Wu et al., 2019; Yu & Qin, 2020). However, relying solely on these low-frequency signals can lead to over-smoothing, where node representations become indistinguishable (Bo et al., 2021; Zhang et al., 2024b; Dong et al., 2025; Sun et al., 2022). To mitigate over-smoothing, recent efforts have proposed to dynamically balance low- and high-frequency components of the graph (Dong et al., 2021; Tang et al., 2025; Bo et al., 2021; Ju et al., 2022; Chang et al., 2021; Sun et al., 2024; Wang et al., 2024a). In parallel, approaches that emphasize low-rank modeling of graph signals and structures have demonstrated greater resilience under noisy conditions (Tang et al., 2024; Yang et al., 2023). Moreover, the attention-based GNN, GFSA (Choi et al., 2024), learns to balance the original attention matrix and its high-order approximation, thereby enriching frequency information and alleviating over-smoothing. HONGAT (Zhang et al., 2024a) explicitly addresses over-smoothing by integrating high-order dependencies and introducing sparsity within the attention mechanism. Whereas GFSA (Choi et al., 2024) and HONGAT (Zhang et al., 2024a) learn the combination weights for different powers of the attention matrix solely by fitting to the training labels, our approach explicitly optimizes the combination weights to minimize the kernel complexity, leading to tighter theoretical generalization bounds and improved robustness.

## 3 PROBLEM SETUP

**Notations.** Let $\mathcal{G} = (\mathcal{V}, \mathcal{E}, \mathbf{X})$ denote an attributed graph with $N$ nodes. The node set is given by $\mathcal{V} = \{v_1, v_2, \ldots, v_N\}$ and the edge set satisfies $\mathcal{E} \subseteq \mathcal{V} \times \mathcal{V}$. The matrix $\mathbf{X} \in \mathbb{R}^{N \times D}$ contains the attribute information of all nodes, where $D$ corresponds to the dimensionality of each node's attributes. The adjacency matrix associated with $\mathcal{G}$ is denoted by $\mathbf{A} \in \{0, 1\}^{N \times N}$. $\tilde{\mathbf{A}} = \mathbf{A} + \mathbf{I}$, and the corresponding diagonal degree matrix is defined as $\tilde{\mathbf{D}}$. $[N]$ refers to the set of integers from 1 to $N$ inclusive. A subset $\mathcal{L} \subseteq [N]$ contains $m$ labeled nodes, and its complement $\mathcal{U} = [N] \setminus \mathcal{L}$ has cardinality $u$. The sets $\mathcal{V}_\mathcal{L}$ and $\mathcal{V}_\mathcal{U}$ represent the collections of labeled and unlabeled nodes, respectively, with $|\mathcal{V}_\mathcal{L}| = m$ and $|\mathcal{V}_\mathcal{U}| = u$. For any vector $\mathbf{u} \in \mathbb{R}^N$, the notation $[\mathbf{u}]_\mathcal{A}$ refers to the subvector composed of entries indexed by $\mathcal{A} \subseteq [N]$. In the case where $\mathbf{u}$ is a matrix, $[\mathbf{u}]_\mathcal{A}$ denotes the submatrix consisting of the rows indexed by $\mathcal{A}$. The Frobenius norm of a matrix is denoted by $\|\cdot\|_\mathrm{F}$, and the $p$-norm of a vector is expressed as $\|\cdot\|_p$.

**Problem Description** In real-world graph datasets, noise commonly arises either in the node attributes or in the labels. This noise can severely undermine the quality of node embeddings learned by standard GCL encoders, thereby degrading the performance of classifiers built on top of them. Our goal is to train a GCL encoder whose node representations remain robust under two transductive node classification settings, one where the labels of $\mathcal{V}_\mathcal{L}$ are corrupted, and another where the input attributes $\mathbf{X}$ contain noise. The node representations learned by a GCL encoder are given by $\mathbf{H}(\boldsymbol{\theta}) = g_{\boldsymbol{\theta}}(\mathbf{X}, \mathbf{A})$, where $g_{\boldsymbol{\theta}}(\cdot)$ denotes the GCL encoder parameterized by $\boldsymbol{\theta}$. In this study, $g_{\boldsymbol{\theta}}$ is instantiated as a two-layer GCN (Kipf & Welling, 2017). The resulting representations, $\mathbf{H}(\boldsymbol{\theta}) = \{\mathbf{H}_1(\boldsymbol{\theta}), \mathbf{H}_2(\boldsymbol{\theta}), \ldots, \mathbf{H}_N(\boldsymbol{\theta})\} \in \mathbb{R}^{N \times d}$, serve as input for the transductive node classification task, for which a linear classifier is first trained on $\mathcal{V}_\mathcal{L}$, and then evaluated for predicting the labels of $\mathcal{V}_\mathcal{U}$. We abbreviate $\mathbf{H}(\boldsymbol{\theta})$ as $\mathbf{H}$ for simplicity of the notations in this paper.

**Preliminary: Prototypical GCL (PGCL).** We adopt a contrastive learning framework to optimize the GCL encoder $g(\cdot)$, a two-layer GCN (Kipf & Welling, 2017). Two augmented graph views, denoted as $G^1 = (\mathbf{X}^1, \mathbf{A}^1)$ and $G^2 = (\mathbf{X}^2, \mathbf{A}^2)$, are created. The resulting node representations are given by $\mathbf{H}^1 = g(\mathbf{X}^1, \mathbf{A}^1)$ and $\mathbf{H}^2 = g(\mathbf{X}^2, \mathbf{A}^2)$. We enhance mutual information between $\mathbf{H}^1$ and $\mathbf{H}^2$ using the InfoNCE loss (Li et al., 2021), and incorporate prototypical contrastive learning (Li et al., 2021; Snell et al., 2017) by aligning node embeddings with $K$-means-derived cluster prototypes, computed as $\mathbf{c}_k = \frac{1}{|S_k|} \sum_{\mathbf{H}_i \in S_k} \mathbf{H}_i$ for every $k$ in $[K]$. The training loss combines node-level and prototype-level objectives, $\mathcal{L}_\mathrm{node}$ and $\mathcal{L}_\mathrm{proto}$, which are computed as $\mathcal{L}_\mathrm{node}(\boldsymbol{\theta}) = -\frac{1}{N} \sum_{i=1}^N \log \frac{s(\mathbf{H}_i^1, \mathbf{H}_i^2)}{s(\mathbf{H}_i^1, \mathbf{H}_i^2) + \sum_{j=1}^N s(\mathbf{H}_i^1, \mathbf{H}_j^2)}$ and $\mathcal{L}_\mathrm{proto}(\boldsymbol{\theta}) = -\frac{1}{N} \sum_{i=1}^N \log \frac{\exp(\mathbf{H}_i \cdot \mathbf{c}_k / \tau)}{\sum_{k=1}^K \exp(\mathbf{H}_i \cdot \mathbf{c}_k / \tau)}$, where $s(\mathbf{H}_i^1, \mathbf{H}_i^2)$ is the cosine similarity between $\mathbf{H}_i^1$ and $\mathbf{H}_i^2$. The overall loss function of the PGCL is $\mathcal{L}_\mathrm{GCL}(\boldsymbol{\theta}) = \mathcal{L}_\mathrm{node}(\boldsymbol{\theta}) + \mathcal{L}_\mathrm{proto}(\boldsymbol{\theta})$, and the training algorithm for the PGCL is summarized in Algorithm 1 in Section D of the appendix.

## 4 METHODS

### 4.1 KERNEL COMPLEXITY REDUCED GCL (KCR-GCL) ENCODER

In order to perform node classification with provable generalization guarantee, we propose a new Kernel Complexity Reduced GCL (KCR-GCL) encoder, which applies a novel KCR self-attention to the node representations $\mathbf{H}$ generated by the PGCL encoder, $g_{\boldsymbol{\theta}}$. The output of the KCR self-attention layer on top of the node features $\mathbf{H}$ is $\mathbf{F} = \mathbf{B}\mathbf{H}$, where $\mathbf{F}$ denotes the attention-transformed features, and $\mathbf{B} \in \mathbb{R}^{N \times N}$ is the attention weight matrix. The gram matrix of the node features is given by $\mathbf{K} = \mathbf{H}\mathbf{H}^\top$. Let $\mathbf{B}_0 = \mathbf{K}/\lambda_1$, where $\lambda_1$ is the largest eigenvalue of $\mathbf{K}$. In our KCR self-attention, the attention matrix is defined as $\mathbf{B} := \sum_{m=1}^M \kappa_m \mathbf{B}_0^m$, where $M \geq 1$ is the maximum degree. The coefficients $\{\kappa_m\}_{m=1}^M$ are computed by $\kappa_m = \frac{\exp(\boldsymbol{\alpha}_m)}{\sum_{j=1}^M \exp(\boldsymbol{\alpha}_j)}$, where $\boldsymbol{\alpha} \in \mathbb{R}^M$ are learnable parameters and $\boldsymbol{\alpha}_m$ is the $m$-th element of $\boldsymbol{\alpha}$. The design of the attention matrix $\mathbf{B} = \sum_{m=1}^M \kappa_m \mathbf{B}_0^m$ in KCR-GCL is inspired by polynomial graph filters widely used in graph signal processing (GSP) for both undirected and directed graphs (Choi et al., 2024; Zhang et al., 2024a; Marques et al., 2020). Each term $\mathbf{B}_0^m$ captures $m$-hop feature propagation over the graph defined by the kernel matrix $\mathbf{B}_0$, and the learnable coefficients $\kappa_m$ determine the relative influence of different neighborhood scales, similar to spectral mixing in generalized graph convolutions (Marques et al., 2020). In addition, the design of the attention matrix $\mathbf{B}$ can also reduce the eigenvalues of the

gram matrix of the attention-transformed features, thus leading to lower kernel complexity. In the KCR-GCL encoder, $\mathbf{F}(\boldsymbol{\alpha}, \boldsymbol{\theta}) = \mathbf{B}(\boldsymbol{\alpha})\mathbf{H}(\boldsymbol{\theta})$. We abbreviate $\mathbf{F}(\boldsymbol{\alpha}, \boldsymbol{\theta})$ and $\mathbf{B}(\boldsymbol{\alpha})$ as $\mathbf{F}$ and $\mathbf{B}$ for simplicity of the notations. The resulting gram matrix of the transformed features $\mathbf{F}$ is $\mathbf{K_F}(\boldsymbol{\theta}, \boldsymbol{\alpha}) = \mathbf{FF}^\top = \mathbf{BKB}$, and $\mathbf{K_F}(\boldsymbol{\theta}, \boldsymbol{\alpha})$ is the learned feature kernel matrix of the KCR-GCL encoder. We also abbreviate $\mathbf{K_F}(\boldsymbol{\theta}, \boldsymbol{\alpha})$ as $\mathbf{K_F}$ for simplicity of the notations in the sequel.

We propose to reduce the TNN of the gram matrix $\mathbf{K_F}$. Let $\left\{\widehat{\lambda}_i\right\}_{i=1}^N$ with $\widehat{\lambda}_1 \geq \widehat{\lambda}_2 \ldots \geq \widehat{\lambda}_{\min\{N,d\}} \geq \widehat{\lambda}_{\min\{N,d\}+1} = \ldots, = 0$ be the eigenvalues of $\mathbf{K_F}$. In order to encourage the features $\mathbf{F}$ or the gram matrix $\mathbf{K_F}$ to be low-rank, we explicitly add the TNN $\|\mathbf{K_F}\|_{r_0} := \sum_{i=r_0+1}^N \widehat{\lambda}_i$ to the loss function of the KCR-GCL encoder. The starting rank $r_0 < \min(N, d)$ is the rank of the gram matrix of the features we aim to obtain with the KCR-GCL encoder, that is, if $\|\mathbf{K_F}\|_{r_0} = 0$, then $\mathrm{rank}(\mathbf{K_F}) = r_0$. The training of the KCR-GCL encoder performs the following optimization,

$$\mathcal{L}_{\text{KCR-GCL}}(\boldsymbol{\theta}, \boldsymbol{\alpha}) = \mathcal{L}_{\text{node}}(\boldsymbol{\theta}) + \mathcal{L}_{\text{proto}}(\boldsymbol{\theta}) + \tau \|\mathbf{K_F}(\boldsymbol{\theta}, \boldsymbol{\alpha})\|_{r_0}, \tag{1}$$

where $\tau$ is a weighting parameter chosen by cross-validation described in Section B.2 of the appendix. In our experiments, we select the TNN rank $r_0$ via standard cross-validation across all graph datasets. As reported in Table 6 in Section B.2 of the appendix, the optimal rank $r_0$ consistently falls within the range of $0.1\min\{N, d\}$ to $0.3\min\{N, d\}$. The reduction of the TNN is also inspired by the reduction of the kernel complexity, which is to be defined later in Section 4.2, leading to provable and sharp generalization error of the linear transductive node classifier using the node representations of the KCR-GCL encoder. Our KCR-GCL encoder also outperforms an ablation study model, Low-Rank GCL without the self-attention matrix $\mathbf{B}$, to be detailed in Section 5.3. Algorithm 2 in Section D of the appendix outlines the training procedure for the KCR-GCL encoder.

**Motivation of Learning Low-Rank Features by the KCR-GCL Encoder.** We investigate how the information from the ground-truth clean labels and the label noise is distributed across different eigenvectors of the feature gram matrix $\mathbf{K_F}$ through an eigen-projection analysis. Let $\tilde{\mathbf{Y}} \in \mathbb{R}^{N \times C}$ denote the clean label matrix without noise. We begin by computing the eigenvectors $\mathbf{U}$ of the gram matrix $\mathbf{K_F}$. The eigen-projection score for the $r$-th eigenvector is then given by $p_r = \frac{1}{C} \sum_{c=1}^C \left\|\mathbf{U}^{(r)\top} \tilde{\mathbf{Y}}^{(c)}\right\|_2^2 / \left\|\tilde{\mathbf{Y}}^{(c)}\right\|_2^2$ for $r \in [N]$, where $C$ is the number of classes, and $\tilde{\mathbf{Y}} \in \{0, 1\}^{N \times C}$ consists of one-hot encoded clean labels. Here $\tilde{\mathbf{Y}}^{(c)}$ refers to the $c$-th column of $\tilde{\mathbf{Y}}$. We define $\mathbf{p} = [p_1, \ldots, p_N] \in \mathbb{R}^N$ as the vector of projection values. In the presence of label noise $\mathbf{N} \in \mathbb{R}^{N \times C}$, the observed label matrix becomes $\mathbf{Y} = \tilde{\mathbf{Y}} + \mathbf{N}$. The projection

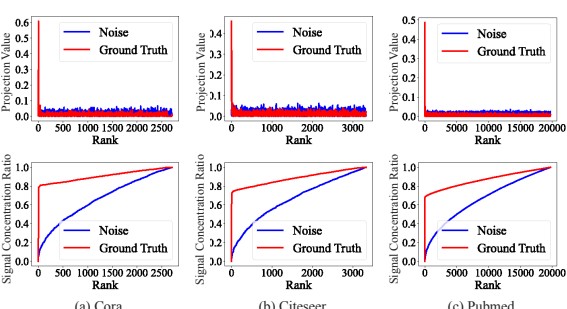

(a) Cora    (b) Citeseer    (c) Pubmed

Figure 1: Eigen-projection (first row) and signal concentration ratio (second row) on Cora, Citeseer, and Pubmed, as the illustration of the Low Frequency Property (LFP). The study in this figure is performed for asymmetric label noise with a noise level of $60\%$. By the rank $r = 0.2\min\{N, d\}$, the signal concentration ratio of $\tilde{\mathbf{Y}}$ for Cora, Citeseer, and Pubmed are $0.844$, $0.809$, and $0.784$ respectively. Figure 2 in Section C.6 of the appendix further illustrates the eigen-projection and signal concentration ratio on more datasets.

value $p_r$ quantifies the proportion of the signal aligned with the $r$-th eigenvector of $\mathbf{K_F}$, while the signal concentration ratio at rank $r$ of the ground truth class label is defined as $\left\|\mathbf{p}^{(1:r)}\right\|_1$, representing the cumulative contribution of the top $r$ eigenvectors. Similarly, the signal concentration ratio at rank $r$ of the noise is defined as $\frac{1}{C} \sum_{c=1}^C \left\|\mathbf{U}^{(r)\top} \mathbf{N}^{(c)}\right\|_2^2 / \|\mathbf{N}^{(c)}\|_2^2 \in \mathbb{R}^N$. Empirical results, shown as red curves in Figure 1, indicate that the clean label signals are primarily concentrated on the leading eigenvectors of $\mathbf{K_F}$. In contrast, the projection of the label noise appears more uniformly distributed across all eigenvectors, as demonstrated by the blue curves in the same figure. The above observation motivates using low-rank features $\mathbf{F}$, or equivalently the low-rank gram matrix $\mathbf{K_F}$, for node classification with label noise. This is because the low-rank part of the feature matrix $\mathbf{F}$ or the gram matrix $\mathbf{K_F}$ covers the dominant information in the ground truth label $\tilde{\mathbf{Y}}$ while learning only a small portion of the label noise. We refer to such property as the **Low Frequency Property**

**(LFP)**, which has been widely studied in deep learning and node-level graph learning (Rahaman et al., 2019; Arora et al., 2019; Cao et al., 2021; Choraria et al., 2022; Wang et al., 2024b; 2025). Moreover, we remark that the regularization term $\|\mathbf{K}_{\mathbf{F}}\|_{r_0}$ in the loss function (1) of KCR-GCL is also theoretically motivated by our sharp upper bound for the test loss using a linear transductive classifier, to be detailed in the next subsection.

### 4.2 Transductive Node Classification With Provable Generalization Guarantee

In this section, we present a linear transductive node classification method based on the node representations $\mathbf{F}$ obtained from the KCR-GCL encoder. For each node $v_i$ where $i \in [N]$, let $\mathbf{y}_i \in \mathbb{R}^C$ denote its observed one-hot class label vector. The classifier's linear prediction is computed by $\mathbf{F}\mathbf{W}$, where $\mathbf{W} \in \mathbb{R}^{d \times C}$ is the learnable weight matrix. Final predictions are made using the softmax transformation $\text{softmax}(\mathbf{F}\mathbf{W}) \in \mathbb{R}^{N \times C}$ to estimate class probabilities for the test nodes. We then train the transductive linear classifier on top of $\mathbf{F}$ by minimizing the loss function,

$$\min_{\mathbf{W}} L(\mathbf{W}, \boldsymbol{\theta}, \boldsymbol{\alpha}) = \frac{1}{m} \sum_{v_i \in \mathcal{V}_\mathcal{L}} \text{KL} \left( \mathbf{y}_i, [\text{softmax}(\mathbf{F}\mathbf{W})]_i \right). \tag{2}$$

We use a regular gradient descent to optimize (2) with a learning rate $\eta \in (0, \frac{1}{\hat{\lambda}_1})$. $\mathbf{W}$ is initialized by $\mathbf{W}^{(0)} = \mathbf{0}$, and at the $t$-th iteration of gradient descent for $t \geq 1$, $\mathbf{W}$ is updated by $\mathbf{W}^{(t)} = \mathbf{W}^{(t-1)} - \eta \nabla_{\mathbf{W}} L(\mathbf{W})|_{\mathbf{W}=\mathbf{W}^{(t-1)}}$. We define $\mathbf{F}(\mathbf{W}, t) \coloneqq \mathbf{F}\mathbf{W}^{(t)}$ as the output of the classifier after the $t$-th iteration of gradient descent for $t \geq 1$. We have the following theoretical result, Theorem 4.1, on the Mean Squared Error (MSE) loss of the unlabeled test nodes $\mathcal{V}_\mathcal{U}$ measured by the gap between $[\mathbf{F}(\mathbf{W}, t)]_\mathcal{U}$ and $\left[\tilde{\mathbf{Y}}\right]_\mathcal{U}$ when using the low-rank feature $\mathbf{F}$ with $r_0 \in [N]$, which is the generalization error bound for the linear transductive classifier using $\mathbf{F}$ to predict the labels of the unlabeled nodes. Similar to existing works such as (Kothapalli et al., 2023) that use the Mean Squared Error (MSE) to analyze the optimization and the generalization of GNNs, we employ the MSE loss to provide the generalization error of the node classifier in the following theorem. It is remarked that the MSE loss is necessary for the generalization analysis of transductive learning using the transductive local Rademacher complexity (Tolstikhin et al., 2014; Yang, 2023; 2025).

**Theorem 4.1.** Let $m \geq cN$ for a constant $c \in (0,1)$, and $r_0 \in [N]$. Assume that a set $\mathcal{L}$ with $|\mathcal{L}| = m$ is sampled uniformly without replacement from $[N]$, and the remaining nodes $\mathcal{V}_U = \mathcal{V} \setminus \mathcal{V}_L$ are the test nodes. Then for every $x > 0$, with probability at least $1 - \exp(-x)$, after the $t$-th iteration of gradient descent for all $t \geq 1$, we have

$$\mathcal{U}_{\text{test}}(t) \coloneqq \frac{1}{u} \left\| \left[ \mathbf{F}(\mathbf{W}, t) - \tilde{\mathbf{Y}} \right]_\mathcal{U} \right\|_\text{F}^2 \leq \frac{2c_0}{m} \left( L_1(\mathbf{K}_\mathbf{F}, \tilde{\mathbf{Y}}, t) + L_2(\mathbf{K}_\mathbf{F}, \mathbf{N}, t) \right) + c_0 \text{KC}(\mathbf{K}_\mathbf{F}) + \frac{c_0 x}{u}, \tag{3}$$

where $c_0$ is a positive number depending on $\mathbf{U}$, $\left\{ \hat{\lambda}_i \right\}_{i=1}^{r_0}$, and $\tau_0$ with $\tau_0^2 = \max_{i \in [N]} [\mathbf{K}_\mathbf{F}]_{ii}$. $L_1(\mathbf{K}_\mathbf{F}, \tilde{\mathbf{Y}}, t) \coloneqq \left\| \left( \mathbf{I}_m - \eta [\mathbf{K}_\mathbf{F}]_{\mathcal{L},\mathcal{L}} \right)^t \left[ \tilde{\mathbf{Y}} \right]_\mathcal{L} \right\|_\text{F}^2$, $L_2(\mathbf{K}_\mathbf{F}, \mathbf{N}, t) = \left\| \eta [\mathbf{K}_\mathbf{F}]_{\mathcal{L},\mathcal{L}} \sum_{t'=0}^{t-1} \left( \mathbf{I}_m - \eta [\mathbf{K}_\mathbf{F}]_{\mathcal{L},\mathcal{L}} \right)^{t'} [\mathbf{N}]_\mathcal{L} \right\|_\text{F}^2$. KC is the kernel complexity of the gram matrix defined by $\text{KC}(\mathbf{K}_\mathbf{F}) = \min_{r_0 \in [N]} r_0 \left( \frac{1}{u} + \frac{1}{m} \right) + \sqrt{\|\mathbf{K}_\mathbf{F}\|_{r_0}} \left( \frac{1}{\sqrt{u}} + \frac{1}{\sqrt{m}} \right)$.

This theorem is proved in Section A of the appendix, and the sharpness of the upper bound for the generalization error, as the RHS of (3), is proved in (Yang, 2025; 2023). Specifically, $\mathcal{U}_{\text{test}}(t)$ denotes the test loss over unlabeled nodes, quantified by the discrepancy between the classifier output $\mathbf{F}(\mathbf{W}, t)$ and the clean label matrix $\tilde{\mathbf{Y}}$. The upper bound on the test loss in (3) consists of three components: $L_1(\mathbf{K}_\mathbf{F}, \tilde{\mathbf{Y}}, t)$, $L_2(\mathbf{K}_\mathbf{F}, \mathbf{N}, t)$, and $\text{KC}(\mathbf{K}_\mathbf{F})$, each serving a distinct role. $L_1(\mathbf{K}_\mathbf{F}, \tilde{\mathbf{Y}}, t)$ reflects the training loss of the classifier when using clean labels. $L_2(\mathbf{K}_\mathbf{F}, \mathbf{N}, t)$ captures the impact of label noise on the classification loss. $\text{KC}(\mathbf{K}_\mathbf{F})$ denotes the kernel complexity (KC) of the gram matrix $\mathbf{K}_\mathbf{F}$.

The design of the self-attention matrix $\mathbf{B}$ is to reduce the eigenvalues of the gram matrix $\mathbf{K}$ of the PGCL. Let $\{\lambda_i\}_{i=1}^N$ denote the eigenvalues of $\mathbf{K}$, ordered as $\lambda_1 \geq \lambda_2 \geq \ldots \geq \lambda_N \geq 0$. Since $\mathbf{B}$ has

the same eigenvectors as $\mathbf{K}$, and the maximum eigenvalue of $\mathbf{B}$ falls in $(0, 1]$, it can be verified that $\widehat{\lambda}_i \leq \lambda_i$. As a result, the KCR self-attention layer reduces the kernel complexity of the original gram matrix $\mathbf{K}$, rendering a sharper upper bound for transductive node classification. This is reflected in Table 3, where the ablation study model, LR-GCL without the KCR self-attention layer, exhibits larger kernel complexity and generalization upper bounds than those of our KCR-GCL. Importantly, the TNN $\|\mathbf{K_F}\|_{r_0}$ appears in the upper bound in (3), thereby providing theoretical justification for incorporating the TNN regularizer $\|\mathbf{K_F}\|_{r_0}$ to promote low-rank feature learning in our KCR-GCL encoder. Furthermore, under the LFP, which is consistently supported by the empirical evidence shown in Figure 1, $L_1(\mathbf{K_F}, \tilde{\mathbf{Y}}, t)$ diminishes as the number of training iterations $t$ increases. Simultaneously, $L_2(\mathbf{K_F}, \mathbf{N}, t)$ remains small due to the approximately uniform eigen-projection of label noise, while $\mathbf{K_F}$ remains close to a rank-$r_0$ matrix, as the TNN is effectively minimized through the KCR-GCL training objective (1). We note that while the theoretical guarantee in Theorem 4.1 is for label noise, the LFP also holds for attribute noise, to be shown in Section C.6, which motivates KCR-GCL for node classification under either noisy labels or noisy attributes in the next section.

# 5 EXPERIMENTS

In this section, we present a thorough evaluation of KCR-GCL across multiple standard graph benchmarks. The experiment settings are detailed in Section 5.1. Performance under semi-supervised node classification settings with different types of label noise is discussed in Section 5.2. We perform an ablation study to verify the effectiveness of the KCR self-attention layer in KCR-GCL in Section 5.3. The kernel complexity (KC) and the theoretical upper bound on test loss for both models are analyzed in Section 5.4, with additional results on KC across more datasets provided in Section C.5. Section 5.5 explores the applicability of our models on heterophilic graphs. Further experimental results can be found in the appendix. Section C.1 of the appendix expands the node classification benchmarks and includes comparisons with more baseline models, while Section C.2 compares our method against existing graph contrastive learning approaches with diverse classifier designs. To evaluate the reliability of the observed gains in Section 5.2 and Section 5.5, we perform Student's $t$-test, with full results reported in Section C.3 of the appendix. The sensitivity analysis on hyperparameters $\tau$, $M$, $r_0$ are conducted in Section C.4. Additional results of eigen-projection visualizations and signal concentration ratios are provided in Section C.6. Lastly, Section C.7 compares the computational efficiency of KCR-GCL with other baselines.

## 5.1 EXPERIMENTAL SETTINGS

We evaluate our proposed approaches on eight widely recognized graph benchmark datasets: Cora, Citeseer, PubMed (Sen et al., 2008), Coauthor CS, ogbn-arxiv (Hu et al., 2020a), Wiki-CS (Mernyei & Cangea, 2020), and the Amazon-Computers and Amazon-Photos datasets (Shchur et al., 2018). Detailed statistics of the datasets are presented in Table 5 in Section B.1 of the appendix. As these datasets do not inherently contain label or feature noise, we synthetically introduce the symmetric and asymmetric label noise following (Han et al., 2020; Dai et al., 2022; Qian et al., 2023), with details in B.4. We simulate attribute noise by randomly permuting a fixed fraction of each node's attributes following (Ding et al., 2022). All experiments utilize the standard train/validation/test splits defined in prior studies (Shchur et al., 2018; Mernyei & Cangea, 2020; Hu et al., 2020a). Noise is only introduced into the training and validation sets to preserve the integrity of the test data for fair performance evaluation. Details on the training settings of KCR-GCL and the cross-validation for selecting the rank parameter $r_0$, the regularization weight $\tau$ associated with the TNN loss, and the value of the maximum power, $M$, are presented in Section B.2 of the appendix.

## 5.2 NODE CLASSIFICATION

To rigorously assess the robustness of KCR-GCL, we conduct extensive experiments on graphs affected by both symmetric and asymmetric label noise, with corruption rates ranging from $40\%$ to $80\%$ in increments of $20\%$. In parallel, we examine the impact of attribute perturbations under the same levels of noise. Details of the compared methods are presented in Section B.3 of the appendix. Table 1 shows the average classification accuracy and standard deviation across 10 runs on the Cora, Citeseer, PubMed, and ogbn-arxiv datasets, comparing KCR-GCL with the strongest baselines. An expanded comparison including additional baselines is provided in Table 7 in Section C.1 of the appendix. Moreover, Table 8 in Section C.1 of the appendix presents additional results on Coauthor-CS, Wiki-CS, Amazon-Computers, and Amazon-Photos under both types of label noise and varying

degrees of attribute noise. It is observed that KCR-GCL consistently achieves the best performance across all datasets and noise levels. For example, under $80\%$ symmetric label noise on PubMed, KCR-GCL outperforms RTGNN, the strongest baseline, by $4.7\%$ in accuracy.

Table 1: Performance comparison against the best-performing baseline methods for node classification on Cora, Citeseer, PubMed, and the large-scale graphs, ogbn-arxiv and Reddit, with asymmetric label noise, symmetric label noise, and attribute noise. Comparisons with more baseline methods on Cora, Citeseer, PubMed, ogbn-arxiv, and Reddit are presented in Table 7 in Section C.1 of the appendix. The highest values for each dataset under each setting are bold. The results are the mean values computed over 10 independent runs, with the standard deviation after $\pm$.

| Dataset | Methods | Noise Type | | | | | | | | | |
| | | 0 | 40 | | | 60 | | | 80 | | |
| | | - | Asymmetric | Symmetric | Attribute | Asymmetric | Symmetric | Attribute | Asymmetric | Symmetric | Attribute |
| --- | --- | --- | --- | --- | --- | --- | --- | --- | --- | --- | --- |
| Cora | GCN | 0.815±0.005 | 0.547±0.015 | 0.636±0.007 | 0.639±0.008 | 0.405±0.014 | 0.517±0.010 | 0.439±0.012 | 0.265±0.012 | 0.354±0.014 | 0.317±0.013 |
| | RTGNN | 0.828±0.003 | 0.570±0.010 | 0.682±0.008 | 0.678±0.011 | 0.474±0.011 | 0.555±0.010 | 0.457±0.009 | 0.280±0.011 | 0.386±0.014 | 0.342±0.016 |
| | MERIT | 0.831±0.005 | 0.560±0.008 | 0.670±0.008 | 0.671±0.009 | 0.467±0.013 | 0.547±0.013 | 0.450±0.014 | 0.277±0.013 | 0.385±0.013 | 0.335±0.009 |
| | ARIEL | 0.843±0.004 | 0.573±0.013 | 0.681±0.010 | 0.675±0.009 | 0.471±0.012 | 0.553±0.012 | 0.455±0.014 | 0.284±0.014 | 0.389±0.013 | 0.343±0.013 |
| | SFA | 0.839±0.010 | 0.564±0.011 | 0.677±0.013 | 0.676±0.015 | 0.473±0.014 | 0.549±0.014 | 0.457±0.014 | 0.282±0.016 | 0.389±0.013 | 0.344±0.017 |
| | GRAND+ | 0.858±0.006 | 0.570±0.009 | 0.682±0.007 | 0.678±0.011 | 0.472±0.010 | 0.554±0.008 | 0.456±0.012 | 0.284±0.015 | 0.387±0.015 | 0.345±0.013 |
| | GFSA | 0.837±0.006 | 0.568±0.012 | 0.676±0.010 | 0.672±0.009 | 0.466±0.012 | 0.545±0.013 | 0.451±0.012 | 0.279±0.012 | 0.384±0.015 | 0.336±0.013 |
| | HONGAT | 0.833±0.004 | 0.566±0.011 | 0.673±0.011 | 0.667±0.010 | 0.464±0.010 | 0.543±0.011 | 0.449±0.010 | 0.278±0.013 | 0.381±0.014 | 0.334±0.014 |
| | CGNN | 0.835±0.006 | 0.567±0.009 | 0.670±0.012 | 0.669±0.011 | 0.467±0.013 | 0.544±0.011 | 0.450±0.014 | 0.281±0.012 | 0.380±0.013 | 0.337±0.014 |
| | KCR-GCL | **0.861±0.006** | **0.610±0.011** | **0.731±0.007** | **0.715±0.011** | **0.512±0.011** | **0.610±0.013** | **0.500±0.012** | **0.341±0.012** | **0.444±0.012** | **0.390±0.011** |
| Citeseer | GCN | 0.703±0.005 | 0.475±0.023 | 0.501±0.013 | 0.529±0.009 | 0.351±0.014 | 0.341±0.014 | 0.372±0.011 | 0.291±0.022 | 0.281±0.019 | 0.290±0.014 |
| | RTGNN | 0.746±0.008 | 0.498±0.007 | 0.556±0.007 | 0.550±0.012 | 0.392±0.010 | 0.424±0.013 | 0.390±0.014 | 0.348±0.017 | 0.308±0.016 | 0.302±0.011 |
| | MERIT | 0.740±0.007 | 0.496±0.012 | 0.536±0.012 | 0.542±0.010 | 0.383±0.011 | 0.425±0.011 | 0.387±0.008 | 0.344±0.014 | 0.301±0.014 | 0.295±0.009 |
| | SFA | 0.740±0.011 | 0.502±0.014 | 0.532±0.015 | 0.547±0.013 | 0.390±0.014 | 0.433±0.014 | 0.389±0.012 | 0.347±0.016 | 0.312±0.015 | 0.299±0.013 |
| | GRAND+ | 0.756±0.004 | 0.497±0.010 | 0.553±0.010 | 0.552±0.011 | 0.390±0.013 | 0.422±0.013 | 0.387±0.013 | 0.348±0.013 | 0.309±0.014 | 0.302±0.012 |
| | GFSA | 0.743±0.006 | 0.495±0.012 | 0.546±0.012 | 0.546±0.011 | 0.386±0.011 | 0.418±0.011 | 0.386±0.012 | 0.342±0.013 | 0.308±0.015 | 0.298±0.012 |
| | HONGAT | 0.738±0.007 | 0.492±0.014 | 0.540±0.011 | 0.545±0.009 | 0.380±0.012 | 0.413±0.010 | 0.384±0.013 | 0.340±0.014 | 0.306±0.016 | 0.296±0.011 |
| | CGNN | 0.741±0.007 | 0.493±0.013 | 0.544±0.012 | 0.546±0.010 | 0.385±0.013 | 0.419±0.012 | 0.385±0.011 | 0.343±0.013 | 0.307±0.013 | 0.297±0.012 |
| | KCR-GCL | **0.761±0.010** | **0.535±0.013** | **0.599±0.013** | **0.588±0.007** | **0.431±0.014** | **0.473±0.014** | **0.425±0.012** | **0.398±0.012** | **0.359±0.014** | **0.341±0.010** |
| PubMed | GCN | 0.790±0.007 | 0.584±0.022 | 0.574±0.012 | 0.595±0.012 | 0.405±0.025 | 0.386±0.011 | 0.488±0.013 | 0.305±0.022 | 0.295±0.013 | 0.423±0.013 |
| | RTGNN | 0.797±0.004 | 0.610±0.008 | 0.622±0.010 | 0.614±0.012 | 0.455±0.010 | 0.455±0.011 | 0.501±0.011 | 0.335±0.013 | 0.338±0.017 | 0.452±0.013 |
| | MERIT | 0.801±0.004 | 0.593±0.011 | 0.612±0.011 | 0.613±0.011 | 0.447±0.012 | 0.443±0.012 | 0.497±0.009 | 0.328±0.011 | 0.323±0.011 | 0.445±0.009 |
| | SFA | 0.804±0.010 | 0.596±0.011 | 0.615±0.011 | 0.609±0.011 | 0.447±0.014 | 0.446±0.017 | 0.499±0.014 | 0.330±0.011 | 0.327±0.014 | 0.447±0.014 |
| | GRAND+ | 0.845±0.006 | 0.610±0.011 | 0.624±0.013 | 0.617±0.013 | 0.453±0.008 | 0.453±0.011 | 0.503±0.010 | 0.331±0.014 | 0.337±0.013 | 0.458±0.014 |
| | GFSA | 0.823±0.005 | 0.608±0.012 | 0.621±0.011 | 0.616±0.009 | 0.450±0.013 | 0.452±0.012 | 0.500±0.010 | 0.333±0.013 | 0.334±0.011 | 0.455±0.012 |
| | HONGAT | 0.818±0.006 | 0.606±0.011 | 0.619±0.012 | 0.613±0.010 | 0.448±0.014 | 0.447±0.012 | 0.498±0.012 | 0.328±0.012 | 0.326±0.013 | 0.450±0.011 |
| | CGNN | 0.822±0.006 | 0.607±0.013 | 0.620±0.011 | 0.615±0.010 | 0.449±0.012 | 0.451±0.014 | 0.499±0.010 | 0.330±0.012 | 0.330±0.012 | 0.454±0.013 |
| | KCR-GCL | **0.846±0.009** | **0.655±0.014** | **0.669±0.015** | **0.653±0.011** | **0.493±0.011** | **0.501±0.013** | **0.544±0.011** | **0.381±0.011** | **0.385±0.012** | **0.502±0.014** |
| ogbn-arxiv | GCN | 0.717±0.003 | 0.401±0.014 | 0.421±0.014 | 0.478±0.010 | 0.336±0.011 | 0.346±0.021 | 0.339±0.012 | 0.286±0.022 | 0.256±0.010 | 0.294±0.013 |
| | RTGNN | 0.718±0.004 | 0.443±0.012 | 0.464±0.012 | 0.484±0.014 | 0.380±0.011 | 0.384±0.013 | 0.340±0.017 | 0.335±0.011 | 0.285±0.015 | 0.301±0.006 |
| | MERIT | 0.717±0.004 | 0.442±0.009 | 0.463±0.009 | 0.483±0.010 | 0.368±0.011 | 0.381±0.011 | 0.341±0.012 | 0.324±0.012 | 0.272±0.010 | 0.304±0.009 |
| | SFA | 0.718±0.009 | 0.445±0.012 | 0.463±0.013 | 0.486±0.012 | 0.368±0.011 | 0.378±0.014 | 0.338±0.015 | 0.325±0.014 | 0.273±0.012 | 0.302±0.013 |
| | GRAND+ | 0.725±0.004 | 0.445±0.008 | 0.466±0.011 | 0.481±0.011 | 0.378±0.010 | 0.385±0.012 | 0.344±0.010 | 0.332±0.010 | 0.282±0.016 | 0.303±0.009 |
| | GFSA | 0.719±0.004 | 0.443±0.012 | 0.460±0.010 | 0.482±0.011 | 0.370±0.012 | 0.379±0.012 | 0.342±0.011 | 0.328±0.012 | 0.278±0.013 | 0.299±0.011 |
| | HONGAT | 0.716±0.005 | 0.440±0.011 | 0.458±0.012 | 0.480±0.012 | 0.366±0.013 | 0.373±0.013 | 0.339±0.012 | 0.324±0.014 | 0.276±0.014 | 0.296±0.012 |
| | CGNN | 0.717±0.006 | 0.441±0.013 | 0.462±0.011 | 0.481±0.010 | 0.368±0.014 | 0.376±0.012 | 0.340±0.011 | 0.326±0.015 | 0.277±0.013 | 0.298±0.012 |
| | KCR-GCL | **0.733±0.006** | **0.491±0.013** | **0.511±0.011** | **0.523±0.014** | **0.423±0.014** | **0.435±0.012** | **0.425±0.012** | **0.379±0.015** | **0.337±0.013** | **0.352±0.013** |
| Reddit | GCN | 0.960±0.003 | 0.543±0.020 | 0.571±0.018 | 0.642±0.018 | 0.438±0.025 | 0.462±0.022 | 0.452±0.020 | 0.384±0.025 | 0.348±0.020 | 0.388±0.020 |
| | RTGNN | 0.962±0.004 | 0.561±0.018 | 0.588±0.017 | 0.661±0.016 | 0.458±0.020 | 0.483±0.020 | 0.471±0.018 | 0.402±0.022 | 0.363±0.019 | 0.409±0.018 |
| | MERIT | 0.961±0.004 | 0.556±0.019 | 0.584±0.018 | 0.653±0.017 | 0.456±0.021 | 0.476±0.021 | 0.467±0.019 | 0.397±0.023 | 0.353±0.020 | 0.407±0.018 |
| | SFA | 0.963±0.005 | 0.559±0.018 | 0.592±0.017 | 0.659±0.016 | 0.459±0.020 | 0.479±0.020 | 0.468±0.018 | 0.401±0.022 | 0.362±0.020 | 0.417±0.018 |
| | GRAND+ | 0.966±0.003 | 0.573±0.017 | 0.603±0.016 | 0.672±0.015 | 0.472±0.019 | 0.488±0.019 | 0.481±0.017 | 0.407±0.021 | 0.367±0.018 | 0.423±0.017 |
| | GFSA | 0.962±0.004 | 0.567±0.018 | 0.593±0.017 | 0.667±0.016 | 0.467±0.020 | 0.486±0.020 | 0.476±0.018 | 0.398±0.022 | 0.361±0.019 | 0.416±0.017 |
| | HONGAT | 0.961±0.004 | 0.562±0.018 | 0.589±0.017 | 0.663±0.016 | 0.461±0.020 | 0.482±0.020 | 0.469±0.018 | 0.396±0.022 | 0.357±0.020 | 0.413±0.018 |
| | CGNN | 0.962±0.005 | 0.563±0.018 | 0.590±0.017 | 0.666±0.016 | 0.465±0.020 | 0.484±0.020 | 0.473±0.018 | 0.398±0.022 | 0.360±0.019 | 0.415±0.018 |
| | KCR-GCL | **0.970±0.003** | **0.600±0.016** | **0.630±0.015** | **0.690±0.014** | **0.500±0.018** | **0.520±0.018** | **0.510±0.016** | **0.420±0.020** | **0.380±0.018** | **0.440±0.017** |

## 5.3 ABLATION STUDY ON THE KCR GRAPH-ATTENTION LAYER

To study the effectiveness of the KCR self-attention layer, we design an ablation model of KCR-GCL, referred to as Low Rank-GCL (LR-GCL), without the KCR self-attention layer. The TNN, $\|\mathbf{K}(\boldsymbol{\theta})\|_{r_0}$, is incorporated as the low-rank regularization term on the kernel gram matrix $\mathbf{K}(\boldsymbol{\theta}) = \mathbf{H}(\boldsymbol{\theta})\mathbf{H}(\boldsymbol{\theta})^\top$ to train the GCL encoder $g_{\boldsymbol{\theta}}$, instead of the KCR-GCL encoder. Compared to KCR-GCL, LR-GCL lacks the polynomial graph filtering or graph convolutions implemented by the KCR self-attention layer. We train the LR-GCL encoder by minimizing $\mathcal{L}_{\text{LR-GCL}}(\boldsymbol{\theta}) = \mathcal{L}_{\text{node}}(\boldsymbol{\theta}) + \mathcal{L}_{\text{proto}}(\boldsymbol{\theta}) + \tau_0\|\mathbf{K}(\boldsymbol{\theta})\|_{r_0}$, where $\tau_0 > 0$ is the weighting parameter for the TNN, $\|\mathbf{K}(\boldsymbol{\theta})\|_{r_0}$, which is decided by the same cross-validation process as described in Section 5.1. The study is performed on four benchmark datasets, Cora, Citeseer, PubMed, and ogbn-arxiv, under various types and levels of noise, following Section 5.2. Table 2 demonstrates that incorporating the KCR self-attention layer consistently improves node classification performance across all datasets and noise settings. For example, on the Cora dataset under $60\%$ symmetric label noise, KCR-GCL outperforms LR-GCL by $2.3\%$, highlighting the effectiveness of the KCR self-attention layer $\mathbf{B}$ with graph convolution, which learns rich frequency information and alleviates over-smoothing, similar to GFSA (Choi et al., 2024) and HONGAT (Zhang et al., 2024a).

## 5.4 STUDY ON THE KERNEL COMPLEXITY AND THE UPPER BOUND OF THE TEST LOSS

We present a detailed comparison of the individual components that constitute the upper bound of the test loss defined in Equation (3), namely $L_1(\mathbf{K_F}, \tilde{\mathbf{Y}}, t)$, $L_2(\mathbf{K_F}, \mathbf{N}, t)$, and the kernel complexity term $\text{KC}(\mathbf{K_F})$, based on node representations obtained from various methods. The corresponding

Table 2: Ablation study on the KCR self-attention layer in KCR-GCL for node classification on Cora, Citeseer, PubMed, and ogbn-arxiv with label noise and attribute noise.

| Dataset | Methods | Noise Type | | | | | | | | | |
|---|---|---|---|---|---|---|---|---|---|---|---|
| | | 0 | 40 | | | 60 | | | 80 | | |
| | | - | Asymmetric | Symmetric | Attribute | Asymmetric | Symmetric | Attribute | Asymmetric | Symmetric | Attribute |
| Cora | LR-GCL | 0.858±0.006 | 0.589±0.011 | 0.713±0.007 | 0.695±0.011 | 0.492±0.011 | 0.587±0.013 | 0.477±0.012 | 0.306±0.012 | 0.419±0.012 | 0.363±0.011 |
| | KCR-GCL | **0.861±0.006** | **0.610±0.011** | **0.731±0.007** | **0.715±0.011** | **0.512±0.011** | **0.610±0.013** | **0.500±0.012** | **0.341±0.012** | **0.444±0.012** | **0.390±0.011** |
| Citeseer | LR-GCL | 0.757±0.010 | 0.520±0.013 | 0.581±0.013 | 0.570±0.007 | 0.410±0.014 | 0.455±0.014 | 0.406±0.012 | 0.369±0.012 | 0.335±0.014 | 0.318±0.010 |
| | KCR-GCL | **0.761±0.010** | **0.535±0.013** | **0.599±0.013** | **0.588±0.007** | **0.431±0.014** | **0.473±0.014** | **0.425±0.012** | **0.398±0.012** | **0.359±0.014** | **0.341±0.010** |
| PubMed | LR-GCL | 0.845±0.009 | 0.637±0.014 | 0.645±0.015 | 0.637±0.011 | 0.479±0.011 | 0.484±0.013 | 0.526±0.011 | 0.356±0.011 | 0.360±0.012 | 0.482±0.014 |
| | KCR-GCL | **0.846±0.009** | **0.655±0.014** | **0.669±0.015** | **0.653±0.011** | **0.493±0.011** | **0.501±0.013** | **0.544±0.011** | **0.381±0.011** | **0.385±0.012** | **0.502±0.014** |
| ogbn-arxiv | LR-GCL | 0.728±0.006 | 0.472±0.013 | 0.492±0.011 | 0.508±0.014 | 0.405±0.014 | 0.411±0.012 | 0.405±0.012 | 0.359±0.015 | 0.307±0.013 | 0.335±0.013 |
| | KCR-GCL | **0.733±0.006** | **0.491±0.013** | **0.511±0.011** | **0.523±0.014** | **0.423±0.014** | **0.435±0.012** | **0.425±0.012** | **0.379±0.015** | **0.337±0.013** | **0.352±0.013** |

evaluation results are summarized in Table 3. All experiments are conducted on the Cora, Citeseer, and PubMed datasets under symmetric label noise with a corruption rate of $40\%$. The results demonstrate that KCR-GCL consistently achieves significantly lower values across all three terms when compared to baseline models. These reductions indicate a stronger capacity for generalization in semi-supervised node classification tasks, even under the presence of label noise. In addition, we further compare the KC of the gram matrix computed from node representations generated by KCR-GCL, and competing baselines across more benchmarks in Section C.5 of the appendix.

Table 3: Comparisons on $L_1(\mathbf{K_F}, \tilde{\mathbf{Y}}, t)$, $L_2(\mathbf{K_F}, \mathbf{N}, t)$, $\mathrm{KC}(\mathbf{K_F})$ and the value of the upper bound of the test loss from Theorem 4.1. The lowest values for each dataset in the table are bold, and the second-lowest values are underlined.

| Datasets | | MERIT | SFA | Jo-SRC | GCN | GFSA | HONGAT | LR-GCL | KCR-GCL |
|---|---|---|---|---|---|---|---|---|---|
| Cora | $L_1$ | 5.24 ±0.49 | 6.04 ±0.23 | 6.50 ±0.34 | 7.38 ±0.12 | 6.44 ±0.01 | 6.38 ± 0.13 | 3.72 ± 0.38 | **3.65** ± 0.38 |
| | $L_2$ | 4.92 ±0.14 | 4.95 ±0.35 | 5.05 ±0.13 | 5.24 ±0.01 | 3.80 ±0.24 | 4.25 ± 0.26 | 2.97 ± 0.45 | **2.72** ± 0.42 |
| | KC | 0.37 ±0.29 | 0.42 ±0.09 | 0.48 ±0.39 | 0.44 ±0.40 | 0.35 ±0.31 | 0.40 ± 0.08 | 0.20 ± 0.02 | **0.18** ± 0.26 |
| | Upper Bound | 10.68±0.14 | 11.59±0.15 | 12.18 ±0.46 | 13.22±0.11 | 10.80±0.22 | 11.25 ± 0.02 | 7.05 ± 0.43 | **6.74** ± 0.32 |
| Citeseer | $L_1$ | 4.72 ±0.42 | 4.85 ±0.28 | 4.92 ±0.23 | 5.10 ±0.40 | 4.54 ±0.46 | 4.69 ± 0.19 | 4.02 ± 0.34 | **3.95** ± 0.21 |
| | $L_2$ | 4.33 ±0.04 | 4.69 ±0.07 | 4.42 ±0.15 | 5.08 ±0.25 | 4.20 ±0.00 | 4.42 ± 0.03 | 3.75 ± 0.17 | **3.60** ± 0.22 |
| | KC | 0.47 ±0.27 | 0.45 ±0.18 | 0.55 ±0.08 | 0.64 ±0.42 | 0.47 ±0.10 | 0.50 ± 0.42 | 0.24 ± 0.18 | **0.21** ± 0.16 |
| | Upper Bound | 9.77 ±0.14 | 10.21±0.28 | 10.17 ±0.34 | 11.07±0.24 | 9.40 ±0.25 | 9.84 ± 0.14 | 8.20 ± 0.14 | **7.97** ± 0.33 |
| PubMed | $L_1$ | 3.97 ±0.29 | 4.02 ±0.08 | 4.11 ±0.14 | 4.35 ±0.06 | 4.26 ±0.12 | 3.95 ± 0.23 | 3.38 ± 0.40 | **3.40** ± 0.38 |
| | $L_2$ | 2.69 ±0.20 | 2.54 ±0.28 | 2.60 ±0.32 | 2.88 ±0.08 | 2.98 ±0.09 | 2.85 ± 0.03 | 2.32 ± 0.10 | **2.26** ± 0.45 |
| | KC | 0.54 ±0.49 | 0.50 ±0.27 | 0.62 ±0.17 | 0.71 ±0.23 | 0.52 ±0.17 | 0.66 ± 0.16 | 0.30 ± 0.29 | **0.28** ± 0.37 |
| | Upper Bound | 7.44 ±0.22 | 7.28 ±0.03 | 7.59 ±0.37 | 8.15 ±0.26 | 7.99 ±0.23 | 7.63 ± 0.14 | 6.25 ± 0.30 | **6.16** ± 0.40 |

## 5.5 EVALUATION ON HETEROPHILIC GRAPHS

We evaluate the performance of KCR-GCL on semi-supervised node classification tasks involving two widely used heterophilic graph benchmarks, Texas and Chameleon (Pei et al., 2020). To begin, we illustrate the LFP on Texas and Chameleon in Figure 3 in Section C.6 of the appendix. We adopt TEDGCN (Yan et al., 2023), a GNN tailored for heterophilic graphs, as the encoder backbone for KCR-GCL. As shown in Table 4, KCR-GCL yields substantial improvements over the base TEDGCN model, demonstrating the benefits of reducing kernel complexity under noisy and heterophilic conditions.

Table 4: Performance comparison for node classification on Texas and Chameleon.

| Dataset | Methods | Noise Type | | | | | | | | | |
|---|---|---|---|---|---|---|---|---|---|---|---|
| | | 0 | 40 | | | 60 | | | 80 | | |
| | | - | Asymmetric | Symmetric | Attribute | Asymmetric | Symmetric | Attribute | Asymmetric | Symmetric | Attribute |
| Texas | TEDGCN | 0.771±0.025 | 0.525±0.023 | 0.528±0.018 | 0.541±0.022 | 0.402±0.016 | 0.418±0.019 | 0.445±0.021 | 0.312±0.015 | 0.328±0.017 | 0.341±0.020 |
| | KCR-GCL | **0.785±0.018** | **0.556±0.016** | **0.563±0.013** | **0.576±0.015** | **0.451±0.012** | **0.452±0.014** | **0.472±0.016** | **0.338±0.010** | **0.367±0.012** | **0.382±0.014** |
| Chameleon | TEDGCN | 0.569±0.009 | 0.382±0.021 | 0.401±0.018 | 0.425±0.020 | 0.298±0.017 | 0.315±0.019 | 0.328±0.022 | 0.225±0.016 | 0.241±0.018 | 0.254±0.021 |
| | KCR-GCL | **0.585±0.008** | **0.412±0.016** | **0.444±0.013** | **0.452±0.014** | **0.341±0.011** | **0.352±0.013** | **0.361±0.015** | **0.262±0.010** | **0.282±0.012** | **0.290±0.014** |

# 6 CONCLUSIONS

This paper introduces Kernel Complexity Reduced Graph Contrastive Learning, or KCR-GCL, which consists of a KCR-GCL encoder that learns robust node representation, which will be used for transductive node classification. The KCR-GCL encoder integrates a novel self-attention mechanism that adaptively combines multiple powers of the feature kernel matrix to balance spectral components and reduce kernel complexity. The encoder is trained within a prototypical graph contrastive learning (GCL) framework, with a truncated nuclear norm (TNN) on the gram matrix of the learned features incorporated as a regularizer. The TNN regularizer encourages the KCR-GCL encoder to learn low-rank representations, motivated by the prevalence of low-frequency components in real-world graphs and the theoretical tightness of generalization bounds in transductive settings. Extensive empirical results across diverse graph benchmarks demonstrate that KCR-GCL exhibits strong robustness and consistently outperforms state-of-the-art in learning effective node representations for transductive node classification under noisy conditions, where graphs are subjected to either label corruption or attribute perturbations.

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

## A  THEORETICAL RESULTS

We present the proof of Theorem 4.1 in this section.

**Proof of Theorem 4.1.** Define $\mathbf{N} := \mathbf{Y} - \tilde{\mathbf{Y}} \in \mathbb{R}^N$ as the label noise. It can be verified that at the $t$-th iteration of gradient descent for $t \geq 1$, we have

$$
\mathbf{W}^{(t)} = \mathbf{W}^{(t-1)} - \eta \left[\mathbf{F}\right]_{\mathcal{L}}^{\top} \left[\mathbf{F}\mathbf{W}^{(t-1)} - \mathbf{Y}\right]_{\mathcal{L}}
$$

$$
= \mathbf{W}^{(t-1)} - \eta \left[\mathbf{F}\right]_{\mathcal{L}}^{\top} \left[\mathbf{F}\mathbf{W}^{(t-1)} - \tilde{\mathbf{Y}}\right]_{\mathcal{L}} + \eta \left[\mathbf{F}\right]_{\mathcal{L}}^{\top} \left[\mathbf{N}\right]_{\mathcal{L}}. \tag{4}
$$

It follows by (4) that

$$
\left[\mathbf{F}\right]_{\mathcal{L}} \mathbf{W}^{(t)} = \left[\mathbf{F}\right]_{\mathcal{L}} \mathbf{W}^{(t-1)} - \eta \mathbf{K}_{\mathcal{L},\mathcal{L}} \left[\mathbf{F}\mathbf{W}^{(t-1)} - \tilde{\mathbf{Y}}\right]_{\mathcal{L}} + \eta \left[\mathbf{K}_{\mathbf{F}}\right]_{\mathcal{L},\mathcal{L}} \left[\mathbf{N}\right]_{\mathcal{L}}, \tag{5}
$$

where $\mathbf{K}_{\mathcal{L},\mathcal{L}} := \left[\mathbf{F}\right]_{\mathcal{L}} \left[\mathbf{F}\right]_{\mathcal{L}}^{\top} \in \mathbb{R}^{m \times m}$. With $\mathbf{F}(\mathbf{W}, t) = \mathbf{F}\mathbf{W}^{(t)}$, it follows by (5) that

$$
\left[\mathbf{F}(\mathbf{W}, t) - \tilde{\mathbf{Y}}\right]_{\mathcal{L}} = \left(\mathbf{I}_m - \eta \left[\mathbf{K}_{\mathbf{F}}\right]_{\mathcal{L},\mathcal{L}}\right) \left[\mathbf{F}(\mathbf{W}, t-1) - \tilde{\mathbf{Y}}\right]_{\mathcal{L}} + \eta \left[\mathbf{K}_{\mathbf{F}}\right]_{\mathcal{L},\mathcal{L}} \left[\mathbf{N}\right]_{\mathcal{L}}.
$$

It follows from the above equality and the recursion that

$$
\left[\mathbf{F}(\mathbf{W}, t) - \tilde{\mathbf{Y}}\right]_{\mathcal{L}} = -\left(\mathbf{I}_m - \eta \left[\mathbf{K}_{\mathbf{F}}\right]_{\mathcal{L},\mathcal{L}}\right)^{t} \left[\tilde{\mathbf{Y}}\right]_{\mathcal{L}} + \eta \left[\mathbf{K}_{\mathbf{F}}\right]_{\mathcal{L},\mathcal{L}} \sum_{t'=0}^{t-1} \left(\mathbf{I}_m - \eta \left[\mathbf{K}_{\mathbf{F}}\right]_{\mathcal{L},\mathcal{L}}\right)^{t'} \left[\mathbf{N}\right]_{\mathcal{L}}
$$
$$\tag{6}$$

We apply the sharp tranductive learning bounds in (Yang, 2023; 2025) to obtain the following bound for the test loss $\frac{1}{u}\left\|\left[\mathbf{F}(\mathbf{W}, t) - \tilde{\mathbf{Y}}\right]_{\mathcal{U}}\right\|_{\mathrm{F}}^{2}$:

$$
\frac{1}{u}\left\|\left[\mathbf{F}(\mathbf{W}, t) - \tilde{\mathbf{Y}}\right]_{\mathcal{U}}\right\|_{\mathrm{F}}^{2} \leq \frac{c_0}{m}\left\|\left[\mathbf{F}(\mathbf{W}, t) - \tilde{\mathbf{Y}}\right]_{\mathcal{L}}\right\|_{\mathrm{F}}^{2} + c_0 \min_{0 \leq Q \leq N} r(u, m, Q) + \frac{c_0 x}{u}, \tag{7}
$$

with

$$
r(u, m, Q) := Q\left(\frac{1}{u} + \frac{1}{m}\right) + \left(\sqrt{\frac{\sum_{q=Q+1}^{N} \widehat{\lambda}_q}{u}} + \sqrt{\frac{\sum_{q=Q+1}^{N} \widehat{\lambda}_q}{m}}\right),
$$

where $c_0$ is a positive constant depending on $\mathbf{U}$, $\left\{\widehat{\lambda}_i\right\}_{i=1}^{r}$, and $\tau_0$ with $\tau_0^2 = \max_{i \in [N]} \lfloor K_{\mathbf{F}} \rfloor_{ii}$.

It follows from (6) and (7) that for every $r_0 \in [N]$, we have

$$
\frac{1}{u}\left\|\left[\mathbf{F}(\mathbf{W}, t) - \tilde{\mathbf{Y}}\right]_{\mathcal{U}}\right\|_{\mathrm{F}}^{2} \leq \frac{c_0}{m}\left\|\left(\mathbf{I}_m - \eta \left[\mathbf{K}_{\mathbf{F}}\right]_{\mathcal{L},\mathcal{L}}\right)^{t} \left[\tilde{\mathbf{Y}}\right]_{\mathcal{L}}\right\|_{\mathrm{F}}^{2}
$$

$$
+ c_0 r_0 \left(\frac{1}{u} + \frac{1}{m}\right) + c_0 \left(\sqrt{\frac{\sum_{q=r_0+1}^{N} \widehat{\lambda}_q}{u}} + \sqrt{\frac{\sum_{q=r_0+1}^{N} \widehat{\lambda}_q}{m}}\right) + \frac{c_0 x}{u}
$$

$$
\overset{①}{\leq} \frac{2c_0}{m}\left\|\left(\mathbf{I}_m - \eta \left[\mathbf{K}_{\mathbf{F}}\right]_{\mathcal{L},\mathcal{L}}\right)^{t} \left[\tilde{\mathbf{Y}}\right]_{\mathcal{L}}\right\|_{\mathrm{F}}^{2} + \frac{2c_0}{m}\left\|\eta \left[\mathbf{K}_{\mathbf{F}}\right]_{\mathcal{L},\mathcal{L}} \sum_{t'=0}^{t-1} \left(\mathbf{I}_m - \eta \left[\mathbf{K}_{\mathbf{F}}\right]_{\mathcal{L},\mathcal{L}}\right)^{t'} \left[\mathbf{N}\right]_{\mathcal{L}}\right\|_{\mathrm{F}}^{2}
$$

$$
+ c_0 r_0 \left(\frac{1}{u} + \frac{1}{m}\right) + c_0 \sqrt{\|\mathbf{K}_{\mathbf{F}}\|_{r_0}} \left(\sqrt{\frac{1}{u}} + \sqrt{\frac{1}{m}}\right) + \frac{c_0 x}{u}, \tag{8}
$$

where ① follows from the Cauchy-Schwarz inequality, (6), and $\sum_{q=r_0+1}^{N} \widehat{\lambda}_q = \|\mathbf{K}_{\mathbf{F}}\|_{r_0}$. Equation (3) in Theorem 4.1 of the main paper then follows directly from (8). $\qquad \square$

## B  ADDITIONAL EXPERIMENT SETTINGS

### B.1  DATASET

We assess the effectiveness of our approach on eight established benchmark datasets commonly employed in node representation learning: Cora, Citeseer, PubMed (Sen et al., 2008), Coauthor CS, ogbn-arxiv (Hu et al., 2020a), Wiki-CS (Mernyei & Cangea, 2020), and Amazon-Computers and Amazon-Photos (Shchur et al., 2018). Among these, Cora, Citeseer, and PubMed are canonical citation networks frequently used in the literature. Coauthor CS represents a co-authorship network among computer science researchers, while ogbn-arxiv is a directed citation graph curated from the Open Graph Benchmark. Wiki-CS captures hyperlink connections between computer science entries on Wikipedia. The Amazon-Computers and Amazon-Photos datasets model product co-purchasing behavior on Amazon.com, where nodes correspond to products and edges indicate frequently bought-together relationships. A summary of the key statistics for all datasets is provided in Table 5.

Table 5: Statistics of the datasets.

| Dataset | Nodes | Edges | Features | Classes |
|---|---|---|---|---|
| Cora | 2,708 | 5,429 | 1,433 | 7 |
| CiteSeer | 3,327 | 4,732 | 3,703 | 6 |
| PubMed | 19,717 | 44,338 | 500 | 3 |
| Coauthor CS | 18,333 | 81,894 | 6,805 | 15 |
| ogbn-arxiv | 169,343 | 1,166,243 | 128 | 40 |
| Reddit | 232,965 | 11,606,919 | 602 | 41 |
| Wiki-CS | 11,701 | 215,863 | 300 | 10 |
| Amazon-Computers | 13,752 | 245,861 | 767 | 10 |
| Amazon-Photos | 7,650 | 119,081 | 745 | 8 |

### B.2  ADDITIONAL DETAILS OF THE TRAINING SETTINGS

Hyperparameters are selected via five-fold cross-validation on the training set of each dataset. We sweep the learning rate over $\{1 \times 10^{-4}, 5 \times 10^{-4}, 1 \times 10^{-3}, 5 \times 10^{-3}, 1 \times 10^{-2}, 3 \times 10^{-2}, 6 \times 10^{-2}, 1 \times 10^{-1}, 5 \times 10^{-1}\}$ and the weight decay from $\{1 \times 10^{-5}, 5 \times 10^{-5}, 1 \times 10^{-4}, 5 \times 10^{-4}, 1 \times 10^{-3}, 5 \times 10^{-3}\}$. Dropout probabilities are selected from $\{0.3, 0.4, 0.5, 0.6, 0.7\}$. The best hyperparameters are identified as those minimizing the validation loss. We train all models using the Adam optimizer for a maximum of 500 epochs, employing early stopping if the validation loss does not improve for 20 consecutive epochs. To account for random initialization effects, each configuration is repeated over 10 independent runs using different random seeds.

**Cross-Validation for Tuning** $r_0$, $\tau$**, and** $M$**.** The rank parameter $r_0$, the regularization weight $\tau$ associated with the TNN loss, and the value of the maximum power, $M$, in KCR attention are selected through cross-validation tailored to each dataset. We define the rank as $r_0 = \lceil \gamma \min\{N, d\} \rceil$, where $\gamma$ represents the rank ratio. The hyperparameter $\gamma$ is searched over the set $\{0.1, 0.2, 0.3, 0.4, 0.5, 0.6, 0.7, 0.8, 0.9\}$, while the TNN weight $\tau$ is chosen from $\{0.05, 0.1, 0.15, 0.2, 0.25, 0.3, 0.35, 0.4, 0.45, 0.5\}$. The value of the maximum power, $M$, in KCR attention is selected from $\{1, 2, 3, 4, 5\}$. All the above parameters are tuned using five-fold cross-validation, conducted on a randomly sampled 20% subset of the training data. The final selected values for each dataset are reported in Table 6.

Table 6: Selected rank ratio $\gamma$ and TNN weight $\lambda$ for each dataset.

| Parameters | Cora | Citeseer | PubMed | Coauthor CS | ogbn-arxiv | Wiki-CS | Computers | Photos |
|---|---|---|---|---|---|---|---|---|
| $\tau$ | 0.10 | 0.10 | 0.10 | 0.20 | 0.10 | 0.25 | 0.20 | 0.20 |
| $\gamma$ | 0.2 | 0.2 | 0.3 | 0.3 | 0.4 | 0.2 | 0.2 | 0.3 |
| $M$ | 3 | 2 | 4 | 3 | 3 | 3 | 4 | 3 |

### B.3 ADDITIONAL DETAILS OF THE COMPARED METHODS

We perform an extensive comparison of KCR-GCL against a broad range of semi-supervised node representation learning baselines. This includes classical methods such as GCN (Kipf & Welling, 2017), GCE (Zhang & Sabuncu, 2018), S$^2$GC (Zhu & Koniusz, 2021), and GRAND+(Feng et al., 2022b). To evaluate performance under label corruption, we also include two specialized baselines designed to address noisy labels, NRGNN(Dai et al., 2021) and RTGNN (Qian et al., 2023). To assess contrastive learning capabilities, we compare KCR-GCL with leading GCL-based methods, including GraphCL (You et al., 2020), MERIT (Jin et al., 2021), SUGRL (Mo et al., 2022), and SFA (Zhang et al., 2023). We further evaluate KCR-GCL against CRGNN (Li et al., 2024b) and CGNN (Yuan et al., 2023), which incorporate contrastive paradigms tailored to noisy label environments. Additionally, we compare KCR-GCL with attention-based GNN architectures, GFSA (Choi et al., 2024) and HONGAT (Zhang et al., 2024a), both of which are designed to integrate information across different frequency components of the graph spectrum. To investigate KCR-GCL's resilience in learning robust representations, we adapt two methods originally introduced in the visual domain, Jo-SRC (Yao et al., 2021) and Sel-CL (Li et al., 2022), to the graph setting. Both Jo-SRC and Sel-CL rely on clean sample selection strategies that are architecture-agnostic and thus transferable to graph domains. Jo-SRC identifies clean training examples using a representation-level selection mechanism based on Jensen-Shannon divergence and strengthens robustness through a consistency regularization term applied to the contrastive loss. In our adaptation, we integrate Jo-SRC's selection and regularization components into the MERIT framework. Specifically, we augment MERIT's contrastive objective with the consistency loss from Jo-SRC and restrict training to samples flagged as clean by the divergence-based selection process. Sel-CL, on the other hand, focuses on constructing contrastive pairs exclusively from confidently labeled nodes, determined by evaluating alignment between feature representations and label propagation using cross-entropy. It filters node pairs whose similarity outperforms a dynamic confidence threshold. In our implementation, we adapt Sel-CL's high-confidence pair selection mechanism into MERIT by selecting reliable contrastive pairs based on representation-level agreement, thereby improving robustness to noisy labels in the graph domain.

### B.4 ADDITIONAL DETAILS OF THE LABEL NOISE

To introduce label noise, we follow established methodologies from the literature (Han et al., 2020; Dai et al., 2022; Qian et al., 2023), adopting (1) Symmetric noise, where each label is replaced by a randomly chosen label from the remaining classes with uniform probability; and (2) Asymmetric noise, in which labels are more likely to be flipped to semantically related classes. We implement the formal noise model described in (Song et al., 2022), where a noise transition matrix $\mathbf{T} \in [0,1]^{C \times C}$ is used, with entries $\mathbf{T}_{ij} := \mathbb{P}(\tilde{y} = j \mid y = i)$ denoting the probability of a clean label $i$ being corrupted into a noisy label $j$. For symmetric corruption at noise rate $\tau$, we define $\mathbf{T}_{ii} = 1 - \tau$ and $\mathbf{T}_{ij} = \frac{\tau}{C-1}$ for all $j \neq i$. In the asymmetric case, $\mathbf{T}_{ii} = 1 - \tau$ while the off-diagonal entries are structured such that $\mathbf{T}_{ij} > \mathbf{T}_{ik}$ for certain pairs $j, k \neq i$, reflecting realistic label confusion patterns.

## C ADDITIONAL EXPERIMENT RESULTS

### C.1 ADDITIONAL NODE CLASSIFICATION RESULTS ON MORE DATASETS

Table 7 presents the node classification results under symmetric label noise, asymmetric label noise, and attribute noise on Cora, Citeseer, Pubmed, and ogbn-arxiv. Table 8 presents the node classification results under symmetric label noise, asymmetric label noise, and attribute noise on Coauthor-CS, Wiki-CS, Amazon-Computers, and Amazon-Photos. The table reports the mean accuracy and standard deviation over 10 independent runs. As shown, both KCR-GCL and its ablation model, LR-GCL, consistently outperform all baseline methods across these benchmark datasets, demonstrating superior robustness to both label and attribute noise.

Table 7: Performance comparison for node classification on Cora, Citeseer, PubMed, and ogbn-arxiv with asymmetric label noise, symmetric label noise, and attribute noise. The highest values for each dataset under each setting in the table are bold, and the second-lowest values are underlined. The results represent the mean values computed over 10 independent runs, with the standard deviation reported after ±.

| Dataset | Methods | Noise Type 0 | 40 Asymmetric | 40 Symmetric | 40 Attribute | 60 Asymmetric | 60 Symmetric | 60 Attribute | 80 Asymmetric | 80 Symmetric | 80 Attribute |
|---|---|---|---|---|---|---|---|---|---|---|---|
| Cora | GCN | 0.815±0.005 | 0.547±0.015 | 0.636±0.007 | 0.639±0.008 | 0.405±0.014 | 0.517±0.010 | 0.439±0.012 | 0.265±0.012 | 0.354±0.014 | 0.317±0.013 |
| | S²GC | 0.835±0.002 | 0.569±0.007 | 0.664±0.007 | 0.661±0.007 | 0.422±0.010 | 0.535±0.010 | 0.454±0.011 | 0.279±0.014 | 0.366±0.014 | 0.320±0.013 |
| | GCE | 0.819±0.004 | 0.573±0.011 | 0.652±0.008 | 0.650±0.014 | 0.449±0.011 | 0.509±0.011 | 0.445±0.015 | 0.280±0.013 | 0.353±0.013 | 0.325±0.015 |
| | UnionNET | 0.820±0.006 | 0.569±0.014 | 0.664±0.007 | 0.653±0.012 | 0.452±0.010 | 0.541±0.010 | 0.450±0.009 | 0.283±0.014 | 0.370±0.011 | 0.320±0.012 |
| | NRGNN | 0.822±0.006 | 0.571±0.019 | 0.676±0.007 | 0.645±0.012 | 0.470±0.014 | 0.548±0.014 | 0.451±0.011 | 0.282±0.022 | 0.373±0.012 | 0.326±0.010 |
| | RTGNN | 0.828±0.003 | 0.570±0.010 | 0.682±0.008 | 0.678±0.011 | 0.474±0.011 | 0.555±0.010 | 0.457±0.009 | 0.280±0.011 | 0.386±0.014 | 0.342±0.016 |
| | SUGRL | 0.834±0.005 | 0.564±0.011 | 0.674±0.012 | 0.675±0.009 | 0.468±0.011 | 0.552±0.011 | 0.452±0.012 | 0.280±0.012 | 0.381±0.012 | 0.338±0.014 |
| | MERIT | 0.831±0.005 | 0.560±0.008 | 0.670±0.008 | 0.671±0.009 | 0.467±0.013 | 0.547±0.013 | 0.450±0.014 | 0.277±0.013 | 0.385±0.013 | 0.335±0.009 |
| | ARIEL | 0.843±0.004 | 0.573±0.013 | 0.681±0.010 | 0.675±0.009 | 0.471±0.012 | 0.553±0.012 | 0.455±0.014 | 0.284±0.014 | 0.389±0.013 | 0.343±0.013 |
| | SFA | 0.839±0.010 | 0.564±0.011 | 0.677±0.013 | 0.676±0.015 | 0.473±0.014 | 0.549±0.014 | 0.457±0.014 | 0.282±0.016 | 0.389±0.013 | 0.344±0.017 |
| | Sel-Cl | 0.828±0.002 | 0.570±0.010 | 0.685±0.012 | 0.676±0.009 | 0.472±0.013 | 0.554±0.014 | 0.455±0.011 | 0.282±0.017 | 0.389±0.013 | 0.341±0.015 |
| | Jo-SRC | 0.825±0.005 | 0.571±0.006 | 0.684±0.013 | 0.679±0.007 | 0.473±0.011 | 0.556±0.008 | 0.458±0.012 | 0.285±0.013 | 0.387±0.018 | 0.345±0.018 |
| | GRAND+ | 0.858±0.006 | 0.570±0.009 | 0.682±0.007 | 0.678±0.011 | 0.472±0.010 | 0.554±0.008 | 0.456±0.012 | 0.284±0.015 | 0.387±0.015 | 0.345±0.013 |
| | GFSA | 0.837±0.006 | 0.568±0.012 | 0.676±0.010 | 0.672±0.009 | 0.466±0.012 | 0.545±0.013 | 0.451±0.012 | 0.279±0.012 | 0.384±0.015 | 0.336±0.013 |
| | HONGAT | 0.833±0.004 | 0.566±0.011 | 0.673±0.011 | 0.667±0.010 | 0.464±0.010 | 0.543±0.011 | 0.449±0.010 | 0.278±0.013 | 0.381±0.014 | 0.334±0.014 |
| | CRGNN | 0.842±0.005 | 0.572±0.010 | 0.678±0.010 | 0.674±0.010 | 0.470±0.012 | 0.551±0.013 | 0.454±0.013 | 0.283±0.014 | 0.386±0.014 | 0.341±0.015 |
| | CGNN | 0.835±0.006 | 0.567±0.009 | 0.670±0.012 | 0.669±0.011 | 0.462±0.013 | 0.544±0.011 | 0.450±0.013 | 0.281±0.012 | 0.380±0.013 | 0.337±0.014 |
| | KCR-GCL | **0.861±0.006** | **0.602±0.011** | **0.724±0.007** | **0.708±0.011** | **0.510±0.011** | **0.605±0.013** | **0.492±0.012** | **0.329±0.012** | **0.436±0.012** | **0.382±0.011** |
| Citeseer | GCN | 0.703±0.005 | 0.475±0.023 | 0.501±0.013 | 0.529±0.009 | 0.351±0.014 | 0.341±0.014 | 0.372±0.011 | 0.291±0.022 | 0.281±0.019 | 0.290±0.014 |
| | S²GC | 0.736±0.005 | 0.488±0.013 | 0.528±0.013 | 0.553±0.008 | 0.363±0.012 | 0.367±0.014 | 0.390±0.013 | 0.304±0.024 | 0.284±0.019 | 0.288±0.011 |
| | GCE | 0.705±0.004 | 0.490±0.016 | 0.512±0.014 | 0.540±0.014 | 0.362±0.015 | 0.352±0.010 | 0.381±0.009 | 0.309±0.012 | 0.285±0.014 | 0.285±0.011 |
| | UnionNET | 0.706±0.006 | 0.499±0.015 | 0.547±0.014 | 0.545±0.013 | 0.379±0.013 | 0.399±0.013 | 0.379±0.012 | 0.322±0.021 | 0.302±0.013 | 0.290±0.012 |
| | NRGNN | 0.710±0.006 | 0.498±0.015 | 0.546±0.015 | 0.538±0.010 | 0.382±0.016 | 0.412±0.016 | 0.377±0.012 | 0.336±0.021 | 0.309±0.018 | 0.284±0.009 |
| | RTGNN | 0.746±0.008 | 0.498±0.007 | 0.556±0.007 | 0.550±0.012 | 0.392±0.010 | 0.424±0.013 | 0.390±0.014 | 0.348±0.017 | 0.308±0.016 | 0.302±0.011 |
| | SUGRL | 0.730±0.005 | 0.493±0.011 | 0.541±0.011 | 0.544±0.010 | 0.376±0.009 | 0.421±0.009 | 0.388±0.009 | 0.339±0.010 | 0.305±0.010 | 0.300±0.009 |
| | MERIT | 0.740±0.007 | 0.496±0.012 | 0.536±0.012 | 0.542±0.010 | 0.383±0.011 | 0.425±0.011 | 0.387±0.008 | 0.344±0.014 | 0.301±0.014 | 0.295±0.009 |
| | SFA | 0.740±0.011 | 0.502±0.014 | 0.532±0.015 | 0.547±0.013 | 0.390±0.014 | 0.433±0.014 | 0.389±0.012 | 0.347±0.016 | 0.312±0.015 | 0.299±0.013 |
| | ARIEL | 0.727±0.007 | 0.500±0.008 | 0.550±0.013 | 0.548±0.008 | 0.391±0.009 | 0.427±0.012 | 0.389±0.014 | 0.349±0.014 | 0.307±0.013 | 0.299±0.013 |
| | Sel-Cl | 0.725±0.008 | 0.499±0.012 | 0.551±0.010 | 0.549±0.008 | 0.389±0.011 | 0.426±0.008 | 0.391±0.020 | 0.350±0.018 | 0.310±0.015 | 0.300±0.017 |
| | Jo-SRC | 0.730±0.005 | 0.500±0.013 | 0.555±0.011 | 0.551±0.011 | 0.394±0.013 | 0.425±0.013 | 0.393±0.013 | 0.351±0.013 | 0.305±0.018 | 0.303±0.013 |
| | GRAND+ | 0.756±0.004 | 0.497±0.010 | 0.553±0.010 | 0.552±0.011 | 0.390±0.013 | 0.422±0.013 | 0.387±0.013 | 0.348±0.013 | 0.309±0.014 | 0.302±0.012 |
| | GFSA | 0.743±0.006 | 0.495±0.012 | 0.546±0.012 | 0.546±0.011 | 0.386±0.011 | 0.418±0.011 | 0.386±0.012 | 0.342±0.013 | 0.308±0.015 | 0.298±0.012 |
| | HONGAT | 0.738±0.007 | 0.492±0.014 | 0.540±0.011 | 0.545±0.009 | 0.380±0.012 | 0.413±0.010 | 0.384±0.013 | 0.340±0.014 | 0.306±0.016 | 0.296±0.011 |
| | CRGNN | 0.751±0.006 | 0.497±0.011 | 0.552±0.010 | 0.549±0.012 | 0.389±0.014 | 0.423±0.013 | 0.388±0.012 | 0.347±0.015 | 0.310±0.014 | 0.301±0.012 |
| | CGNN | 0.741±0.007 | 0.493±0.013 | 0.544±0.012 | 0.546±0.010 | 0.385±0.013 | 0.419±0.012 | 0.385±0.011 | 0.343±0.013 | 0.307±0.013 | 0.297±0.012 |
| | KCR-GCL | **0.762±0.010** | **0.533±0.013** | **0.597±0.013** | **0.588±0.007** | **0.430±0.014** | **0.472±0.014** | **0.423±0.012** | **0.392±0.012** | **0.352±0.014** | **0.335±0.010** |
| PubMed | GCN | 0.790±0.007 | 0.584±0.022 | 0.574±0.012 | 0.595±0.022 | 0.405±0.025 | 0.386±0.011 | 0.488±0.013 | 0.305±0.022 | 0.295±0.013 | 0.423±0.013 |
| | S²GC | 0.802±0.005 | 0.585±0.023 | 0.589±0.013 | 0.610±0.009 | 0.421±0.030 | 0.401±0.014 | 0.497±0.012 | 0.310±0.039 | 0.290±0.019 | 0.431±0.010 |
| | GCE | 0.792±0.009 | 0.589±0.018 | 0.581±0.011 | 0.590±0.014 | 0.430±0.012 | 0.399±0.012 | 0.491±0.010 | 0.311±0.021 | 0.301±0.011 | 0.424±0.012 |
| | UnionNET | 0.793±0.008 | 0.603±0.020 | 0.620±0.012 | 0.592±0.012 | 0.445±0.022 | 0.424±0.013 | 0.489±0.015 | 0.313±0.025 | 0.327±0.015 | 0.435±0.009 |
| | NRGNN | 0.797±0.008 | 0.602±0.022 | 0.618±0.013 | 0.603±0.008 | 0.443±0.012 | 0.434±0.012 | 0.499±0.009 | 0.330±0.023 | 0.325±0.013 | 0.433±0.011 |
| | RTGNN | 0.797±0.004 | 0.610±0.008 | 0.622±0.010 | 0.614±0.012 | 0.455±0.010 | 0.455±0.011 | 0.501±0.011 | 0.335±0.013 | 0.338±0.017 | 0.452±0.013 |
| | SUGRL | 0.819±0.005 | 0.603±0.013 | 0.615±0.013 | 0.615±0.010 | 0.445±0.011 | 0.441±0.011 | 0.501±0.007 | 0.321±0.009 | 0.321±0.009 | 0.446±0.010 |
| | MERIT | 0.801±0.004 | 0.593±0.011 | 0.612±0.011 | 0.613±0.011 | 0.447±0.012 | 0.443±0.012 | 0.497±0.009 | 0.328±0.011 | 0.323±0.011 | 0.445±0.009 |
| | ARIEL | 0.800±0.003 | 0.610±0.013 | 0.622±0.010 | 0.615±0.011 | 0.453±0.012 | 0.453±0.012 | 0.502±0.014 | 0.331±0.014 | 0.336±0.018 | 0.457±0.013 |
| | SFA | 0.804±0.010 | 0.596±0.011 | 0.615±0.011 | 0.609±0.011 | 0.447±0.014 | 0.446±0.017 | 0.499±0.014 | 0.330±0.011 | 0.327±0.011 | 0.447±0.014 |
| | Sel-Cl | 0.799±0.005 | 0.605±0.014 | 0.625±0.012 | 0.614±0.012 | 0.455±0.014 | 0.449±0.010 | 0.502±0.008 | 0.334±0.021 | 0.332±0.014 | 0.456±0.014 |
| | Jo-SRC | 0.801±0.005 | 0.613±0.010 | 0.624±0.013 | 0.617±0.013 | 0.453±0.008 | 0.455±0.013 | 0.504±0.013 | 0.330±0.015 | 0.334±0.018 | 0.459±0.014 |
| | GRAND+ | 0.845±0.006 | 0.610±0.011 | 0.624±0.013 | 0.617±0.013 | 0.453±0.008 | 0.453±0.011 | 0.503±0.010 | 0.331±0.014 | 0.337±0.013 | 0.458±0.014 |
| | GFSA | 0.823±0.005 | 0.608±0.012 | 0.621±0.011 | 0.616±0.009 | 0.450±0.013 | 0.452±0.012 | 0.500±0.010 | 0.333±0.013 | 0.334±0.011 | 0.455±0.012 |
| | HONGAT | 0.818±0.006 | 0.606±0.011 | 0.619±0.012 | 0.613±0.010 | 0.448±0.014 | 0.447±0.012 | 0.498±0.012 | 0.328±0.012 | 0.326±0.013 | 0.450±0.011 |
| | CRGNN | 0.829±0.005 | 0.612±0.010 | 0.623±0.009 | 0.618±0.011 | 0.452±0.011 | 0.455±0.013 | 0.503±0.009 | 0.335±0.013 | 0.333±0.014 | 0.457±0.012 |
| | CGNN | 0.822±0.006 | 0.607±0.012 | 0.620±0.011 | 0.615±0.010 | 0.449±0.012 | 0.451±0.014 | 0.499±0.010 | 0.332±0.014 | 0.330±0.012 | 0.454±0.013 |
| | KCR-GCL | **0.846±0.009** | **0.652±0.014** | **0.662±0.015** | **0.655±0.011** | **0.498±0.011** | **0.503±0.013** | **0.544±0.011** | **0.379±0.011** | **0.379±0.012** | **0.498±0.014** |
| ogbn-arxiv | GCN | 0.717±0.003 | 0.401±0.014 | 0.421±0.014 | 0.478±0.010 | 0.336±0.011 | 0.346±0.011 | 0.339±0.012 | 0.286±0.022 | 0.256±0.010 | 0.294±0.013 |
| | S²GC | 0.712±0.003 | 0.417±0.017 | 0.429±0.014 | 0.492±0.010 | 0.344±0.016 | 0.353±0.031 | 0.343±0.009 | 0.297±0.023 | 0.266±0.013 | 0.284±0.012 |
| | GCE | 0.720±0.004 | 0.410±0.018 | 0.428±0.008 | 0.480±0.014 | 0.348±0.019 | 0.344±0.019 | 0.342±0.015 | 0.310±0.014 | 0.260±0.011 | 0.275±0.015 |
| | UnionNET | 0.724±0.006 | 0.429±0.021 | 0.449±0.007 | 0.485±0.012 | 0.362±0.018 | 0.367±0.008 | 0.340±0.009 | 0.332±0.019 | 0.269±0.013 | 0.280±0.012 |
| | NRGNN | 0.721±0.006 | 0.449±0.014 | 0.466±0.009 | 0.485±0.012 | 0.371±0.020 | 0.379±0.008 | 0.342±0.011 | 0.330±0.018 | 0.271±0.018 | 0.300±0.010 |
| | RTGNN | 0.718±0.004 | 0.443±0.012 | 0.464±0.012 | 0.484±0.014 | 0.380±0.011 | 0.384±0.013 | 0.340±0.017 | 0.335±0.011 | 0.285±0.015 | 0.301±0.006 |
| | SUGRL | 0.693±0.002 | 0.439±0.010 | 0.467±0.010 | 0.480±0.012 | 0.365±0.013 | 0.385±0.011 | 0.341±0.009 | 0.327±0.011 | 0.275±0.011 | 0.295±0.011 |
| | MERIT | 0.717±0.004 | 0.442±0.009 | 0.463±0.009 | 0.483±0.010 | 0.368±0.011 | 0.381±0.011 | 0.341±0.012 | 0.324±0.012 | 0.272±0.010 | 0.304±0.009 |
| | ARIEL | 0.717±0.004 | 0.448±0.013 | 0.471±0.013 | 0.482±0.011 | 0.379±0.014 | 0.384±0.015 | 0.342±0.015 | 0.334±0.014 | 0.280±0.013 | 0.300±0.010 |
| | SFA | 0.718±0.009 | 0.445±0.012 | 0.463±0.013 | 0.486±0.012 | 0.368±0.011 | 0.378±0.014 | 0.338±0.015 | 0.325±0.014 | 0.273±0.012 | 0.302±0.013 |
| | Sel-Cl | 0.719±0.002 | 0.447±0.007 | 0.469±0.007 | 0.486±0.010 | 0.375±0.008 | 0.389±0.025 | 0.344±0.013 | 0.331±0.008 | 0.284±0.019 | 0.304±0.012 |
| | Jo-SRC | 0.715±0.005 | 0.445±0.011 | 0.466±0.009 | 0.481±0.010 | 0.377±0.013 | 0.387±0.013 | 0.340±0.013 | 0.333±0.013 | 0.282±0.018 | 0.297±0.009 |
| | GRAND+ | 0.725±0.004 | 0.445±0.008 | 0.466±0.011 | 0.481±0.011 | 0.378±0.010 | 0.385±0.012 | 0.344±0.010 | 0.332±0.010 | 0.282±0.016 | 0.303±0.009 |
| | GFSA | 0.719±0.004 | 0.443±0.012 | 0.460±0.010 | 0.482±0.011 | 0.370±0.012 | 0.379±0.012 | 0.342±0.011 | 0.328±0.012 | 0.278±0.013 | 0.299±0.011 |
| | HONGAT | 0.716±0.005 | 0.440±0.011 | 0.458±0.012 | 0.480±0.012 | 0.366±0.013 | 0.373±0.013 | 0.339±0.012 | 0.324±0.014 | 0.276±0.014 | 0.296±0.012 |
| | CRGNN | 0.721±0.003 | 0.446±0.010 | 0.465±0.010 | 0.483±0.009 | 0.372±0.010 | 0.382±0.011 | 0.343±0.010 | 0.330±0.012 | 0.281±0.012 | 0.302±0.010 |
| | CGNN | 0.717±0.006 | 0.441±0.013 | 0.462±0.011 | 0.481±0.010 | 0.368±0.014 | 0.376±0.012 | 0.340±0.011 | 0.326±0.015 | 0.277±0.013 | 0.298±0.012 |
| | KCR-GCL | **0.731±0.006** | **0.487±0.013** | **0.507±0.011** | **0.523±0.014** | **0.423±0.014** | **0.430±0.012** | **0.423±0.012** | **0.374±0.015** | **0.332±0.013** | **0.350±0.013** |

## C.2 NODE CLASSIFICATION RESULTS FOR GCL METHODS WITH DIFFERENT TYPES OF CLASSIFIERS

Existing GCL approaches, including MERIT (Jin et al., 2021), SUGRL (Mo et al., 2022), and SFA (Zhang et al., 2023), typically follow a two-stage procedure: they first train a graph encoder using contrastive objectives such as InfoNCE (Jin et al., 2021), and subsequently train a linear classifier in a supervised manner on the resulting node representations. In contrast, KCR-GCL integrates a transductive classifier directly atop the contrastively learned representations, enabling label propagation during training. To ensure a fair comparison, we retrain all baseline GCL methods using the same transductive classifier employed in KCR-GCL, as well as an additional two-layer transductive GCN classifier. The results with different types of classifiers are shown in Table 9. It is observed

Table 8: Performance comparison for node classification on Coauthor-CS, Wiki-CS, Amazon-Computers, and Amazon-Photos with asymmetric label noise, symmetric label noise, and attribute noise.

| Dataset | Methods | Noise Type | | | | | | | | | |
| | | 0 | 40 | | | 60 | | | 80 | | |
| | | - | Asymmetric | Symmetric | Attribute | Asymmetric | Symmetric | Attribute | Asymmetric | Symmetric | Attribute |
|---|---|---|---|---|---|---|---|---|---|---|---|
| Coauthor-CS | GCN | 0.918±0.001 | 0.645±0.009 | 0.656±0.006 | 0.702±0.010 | 0.511±0.013 | 0.501±0.009 | 0.531±0.010 | 0.429±0.022 | 0.389±0.011 | 0.415±0.013 |
| | S²GC | 0.918±0.001 | 0.657±0.012 | 0.663±0.006 | 0.713±0.010 | 0.516±0.013 | 0.514±0.009 | 0.556±0.009 | 0.437±0.020 | 0.396±0.010 | 0.422±0.012 |
| | GCE | 0.922±0.003 | 0.662±0.017 | 0.659±0.007 | 0.705±0.014 | 0.515±0.016 | 0.502±0.007 | 0.539±0.009 | 0.443±0.017 | 0.389±0.012 | 0.412±0.011 |
| | UnionNET | 0.918±0.002 | 0.669±0.023 | 0.671±0.013 | 0.706±0.012 | 0.525±0.011 | 0.529±0.011 | 0.540±0.012 | 0.458±0.015 | 0.401±0.011 | 0.420±0.007 |
| | NRGNN | 0.919±0.002 | 0.678±0.014 | 0.689±0.009 | 0.705±0.012 | 0.545±0.021 | 0.556±0.011 | 0.546±0.011 | 0.461±0.012 | 0.410±0.012 | 0.417±0.007 |
| | RTGNN | 0.920±0.005 | 0.678±0.012 | 0.691±0.009 | 0.712±0.008 | 0.559±0.010 | 0.569±0.011 | 0.560±0.008 | 0.455±0.005 | 0.415±0.015 | 0.412±0.014 |
| | SUGRL | 0.922±0.005 | 0.675±0.010 | 0.695±0.010 | 0.714±0.006 | 0.550±0.011 | 0.560±0.011 | 0.561±0.007 | 0.449±0.011 | 0.411±0.011 | 0.429±0.008 |
| | MERIT | 0.924±0.004 | 0.679±0.011 | 0.689±0.008 | 0.709±0.005 | 0.552±0.014 | 0.562±0.014 | 0.562±0.011 | 0.452±0.013 | 0.403±0.013 | 0.426±0.005 |
| | ARIEL | 0.925±0.004 | 0.682±0.011 | 0.699±0.009 | 0.712±0.005 | 0.555±0.013 | 0.566±0.012 | 0.556±0.011 | 0.454±0.014 | 0.415±0.019 | 0.427±0.013 |
| | SFA | 0.925±0.009 | 0.682±0.011 | 0.690±0.012 | 0.715±0.012 | 0.555±0.015 | 0.567±0.014 | 0.565±0.013 | 0.458±0.013 | 0.402±0.013 | 0.429±0.015 |
| | Sel-Cl | 0.922±0.008 | 0.684±0.009 | 0.694±0.012 | 0.714±0.010 | 0.557±0.013 | 0.568±0.013 | 0.566±0.010 | 0.457±0.013 | 0.412±0.017 | 0.425±0.009 |
| | Jo-SRC | 0.921±0.005 | 0.684±0.011 | 0.695±0.004 | 0.709±0.007 | 0.560±0.011 | 0.566±0.011 | 0.561±0.009 | 0.456±0.013 | 0.410±0.018 | 0.428±0.010 |
| | GRAND+ | 0.927±0.004 | 0.682±0.011 | 0.693±0.006 | 0.715±0.008 | 0.554±0.008 | 0.568±0.013 | 0.557±0.011 | 0.455±0.012 | 0.416±0.014 | 0.428±0.011 |
| | GFSA | 0.923±0.004 | 0.679±0.010 | 0.687±0.009 | 0.711±0.009 | 0.550±0.012 | 0.559±0.011 | 0.558±0.010 | 0.453±0.014 | 0.410±0.012 | 0.426±0.011 |
| | HONGAT | 0.924±0.003 | 0.681±0.012 | 0.692±0.010 | 0.713±0.008 | 0.553±0.013 | 0.563±0.013 | 0.560±0.012 | 0.456±0.013 | 0.411±0.015 | 0.427±0.010 |
| | CRGNN | 0.926±0.005 | 0.683±0.011 | 0.690±0.011 | 0.712±0.007 | 0.551±0.015 | 0.561±0.012 | 0.559±0.011 | 0.452±0.014 | 0.412±0.014 | 0.426±0.012 |
| | CGNN | 0.925±0.006 | 0.680±0.012 | 0.689±0.012 | 0.710±0.010 | 0.549±0.014 | 0.560±0.012 | 0.557±0.012 | 0.452±0.013 | 0.409±0.015 | 0.425±0.012 |
| | **KCR-GCL** | **0.934±0.006** | **0.714±0.015** | **0.736±0.011** | **0.758±0.015** | **0.594±0.014** | **0.612±0.018** | **0.606±0.015** | **0.489±0.015** | **0.453±0.015** | **0.470±0.017** |
| Wiki-CS | GCN | 0.801±0.004 | 0.612±0.008 | 0.625±0.010 | 0.647±0.009 | 0.497±0.013 | 0.483±0.012 | 0.502±0.011 | 0.401±0.016 | 0.365±0.017 | 0.382±0.016 |
| | S²GC | 0.806±0.003 | 0.621±0.009 | 0.630±0.011 | 0.659±0.010 | 0.503±0.014 | 0.492±0.013 | 0.509±0.012 | 0.411±0.018 | 0.373±0.016 | 0.397±0.015 |
| | GCE | 0.808±0.004 | 0.618±0.007 | 0.629±0.008 | 0.651±0.010 | 0.495±0.012 | 0.481±0.010 | 0.510±0.010 | 0.404±0.015 | 0.361±0.013 | 0.383±0.014 |
| | UnionNET | 0.805±0.005 | 0.629±0.011 | 0.634±0.012 | 0.661±0.009 | 0.506±0.011 | 0.505±0.011 | 0.520±0.012 | 0.421±0.017 | 0.375±0.015 | 0.392±0.014 |
| | NRGNN | 0.809±0.003 | 0.635±0.008 | 0.642±0.009 | 0.665±0.008 | 0.514±0.012 | 0.518±0.011 | 0.526±0.010 | 0.426±0.014 | 0.386±0.016 | 0.403±0.012 |
| | RTGNN | 0.811±0.004 | 0.638±0.010 | 0.645±0.010 | 0.667±0.008 | 0.517±0.011 | 0.522±0.011 | 0.528±0.009 | 0.428±0.013 | 0.391±0.015 | 0.406±0.012 |
| | MERIT | 0.813±0.004 | 0.641±0.009 | 0.648±0.010 | 0.670±0.009 | 0.519±0.012 | 0.525±0.012 | 0.532±0.010 | 0.432±0.014 | 0.392±0.013 | 0.410±0.013 |
| | ARIEL | 0.814±0.003 | 0.645±0.010 | 0.652±0.009 | 0.674±0.008 | 0.523±0.013 | 0.528±0.012 | 0.535±0.011 | 0.434±0.012 | 0.394±0.012 | 0.412±0.012 |
| | SFA | 0.815±0.005 | 0.643±0.011 | 0.650±0.010 | 0.673±0.009 | 0.520±0.013 | 0.527±0.012 | 0.533±0.010 | 0.430±0.015 | 0.391±0.014 | 0.408±0.012 |
| | Sel-Cl | 0.813±0.004 | 0.644±0.009 | 0.651±0.010 | 0.672±0.009 | 0.521±0.012 | 0.526±0.011 | 0.531±0.009 | 0.429±0.013 | 0.390±0.014 | 0.407±0.013 |
| | Jo-SRC | 0.812±0.004 | 0.646±0.010 | 0.652±0.008 | 0.671±0.009 | 0.522±0.012 | 0.528±0.011 | 0.534±0.011 | 0.431±0.014 | 0.393±0.015 | 0.409±0.012 |
| | GRAND+ | 0.816±0.003 | 0.647±0.011 | 0.653±0.009 | 0.676±0.008 | 0.524±0.010 | 0.529±0.011 | 0.536±0.012 | 0.432±0.014 | 0.395±0.013 | 0.411±0.011 |
| | CGNN | 0.813±0.004 | 0.643±0.010 | 0.649±0.009 | 0.669±0.009 | 0.519±0.011 | 0.524±0.011 | 0.531±0.010 | 0.428±0.014 | 0.389±0.015 | 0.406±0.012 |
| | CRGNN | 0.815±0.005 | 0.645±0.010 | 0.652±0.010 | 0.671±0.009 | 0.522±0.013 | 0.528±0.012 | 0.533±0.011 | 0.431±0.013 | 0.392±0.013 | 0.410±0.011 |
| | HONGAT | 0.814±0.004 | 0.642±0.011 | 0.648±0.010 | 0.670±0.009 | 0.518±0.012 | 0.523±0.011 | 0.530±0.010 | 0.427±0.013 | 0.388±0.013 | 0.405±0.012 |
| | **KCR-GCL** | **0.826±0.004** | **0.678±0.013** | **0.699±0.010** | **0.707±0.012** | **0.553±0.014** | **0.572±0.013** | **0.569±0.011** | **0.459±0.014** | **0.426±0.014** | **0.443±0.012** |
| Amazon-Computers | GCN | 0.872±0.005 | 0.619±0.012 | 0.638±0.011 | 0.658±0.013 | 0.471±0.014 | 0.484±0.012 | 0.501±0.010 | 0.377±0.017 | 0.354±0.016 | 0.368±0.015 |
| | S²GC | 0.876±0.004 | 0.625±0.010 | 0.642±0.012 | 0.664±0.011 | 0.479±0.013 | 0.491±0.013 | 0.509±0.012 | 0.382±0.016 | 0.359±0.015 | 0.375±0.014 |
| | GCE | 0.879±0.006 | 0.623±0.011 | 0.641±0.010 | 0.661±0.012 | 0.475±0.014 | 0.486±0.012 | 0.505±0.012 | 0.380±0.015 | 0.356±0.016 | 0.370±0.014 |
| | UnionNET | 0.874±0.005 | 0.633±0.012 | 0.648±0.011 | 0.668±0.010 | 0.483±0.011 | 0.495±0.010 | 0.511±0.011 | 0.388±0.015 | 0.361±0.014 | 0.378±0.013 |
| | NRGNN | 0.878±0.004 | 0.639±0.010 | 0.656±0.010 | 0.672±0.011 | 0.491±0.013 | 0.503±0.011 | 0.518±0.011 | 0.391±0.014 | 0.364±0.015 | 0.380±0.014 |
| | RTGNN | 0.880±0.005 | 0.641±0.010 | 0.658±0.010 | 0.674±0.009 | 0.494±0.012 | 0.507±0.012 | 0.521±0.010 | 0.392±0.014 | 0.366±0.013 | 0.383±0.012 |
| | MERIT | 0.883±0.004 | 0.644±0.011 | 0.660±0.010 | 0.676±0.009 | 0.496±0.012 | 0.508±0.012 | 0.523±0.011 | 0.394±0.015 | 0.368±0.013 | 0.386±0.012 |
| | ARIEL | 0.884±0.004 | 0.645±0.010 | 0.662±0.009 | 0.679±0.010 | 0.498±0.013 | 0.510±0.012 | 0.526±0.011 | 0.396±0.013 | 0.369±0.014 | 0.388±0.012 |
| | SFA | 0.885±0.005 | 0.643±0.010 | 0.661±0.010 | 0.677±0.010 | 0.497±0.012 | 0.509±0.011 | 0.525±0.012 | 0.395±0.013 | 0.368±0.012 | 0.387±0.013 |
| | Sel-Cl | 0.882±0.006 | 0.646±0.009 | 0.663±0.011 | 0.678±0.010 | 0.499±0.012 | 0.511±0.011 | 0.527±0.012 | 0.396±0.013 | 0.369±0.013 | 0.389±0.012 |
| | Jo-SRC | 0.881±0.004 | 0.644±0.010 | 0.661±0.009 | 0.675±0.009 | 0.495±0.011 | 0.508±0.010 | 0.523±0.011 | 0.393±0.014 | 0.367±0.013 | 0.385±0.013 |
| | GRAND+ | 0.886±0.004 | 0.647±0.009 | 0.665±0.009 | 0.680±0.010 | 0.501±0.010 | 0.513±0.010 | 0.529±0.010 | 0.398±0.013 | 0.370±0.013 | 0.390±0.011 |
| | CGNN | 0.884±0.005 | 0.642±0.010 | 0.659±0.010 | 0.676±0.010 | 0.494±0.011 | 0.507±0.011 | 0.522±0.011 | 0.392±0.014 | 0.366±0.013 | 0.384±0.013 |
| | CRGNN | 0.885±0.004 | 0.644±0.009 | 0.662±0.009 | 0.678±0.009 | 0.496±0.012 | 0.509±0.011 | 0.524±0.011 | 0.395±0.013 | 0.368±0.012 | 0.387±0.011 |
| | HONGAT | 0.883±0.005 | 0.640±0.010 | 0.658±0.010 | 0.674±0.010 | 0.492±0.011 | 0.505±0.010 | 0.520±0.010 | 0.392±0.013 | 0.364±0.013 | 0.382±0.012 |
| | **KCR-GCL** | **0.896±0.005** | **0.676±0.014** | **0.694±0.011** | **0.701±0.013** | **0.534±0.014** | **0.548±0.013** | **0.545±0.013** | **0.432±0.014** | **0.401±0.015** | **0.418±0.014** |
| Amazon-Photos | GCN | 0.899±0.004 | 0.638±0.011 | 0.649±0.009 | 0.665±0.011 | 0.487±0.012 | 0.498±0.011 | 0.509±0.011 | 0.395±0.014 | 0.361±0.013 | 0.374±0.012 |
| | S²GC | 0.903±0.005 | 0.645±0.010 | 0.655±0.010 | 0.672±0.010 | 0.495±0.011 | 0.506±0.010 | 0.517±0.011 | 0.399±0.014 | 0.366±0.013 | 0.379±0.013 |
| | GCE | 0.905±0.004 | 0.642±0.011 | 0.654±0.009 | 0.670±0.010 | 0.492±0.012 | 0.503±0.011 | 0.513±0.011 | 0.397±0.013 | 0.364±0.012 | 0.377±0.012 |
| | UnionNET | 0.902±0.004 | 0.648±0.010 | 0.659±0.010 | 0.676±0.010 | 0.497±0.010 | 0.509±0.010 | 0.521±0.011 | 0.403±0.013 | 0.368±0.012 | 0.381±0.011 |
| | NRGNN | 0.906±0.003 | 0.653±0.010 | 0.663±0.010 | 0.680±0.010 | 0.503±0.011 | 0.514±0.010 | 0.526±0.010 | 0.408±0.013 | 0.371±0.012 | 0.386±0.012 |
| | RTGNN | 0.908±0.004 | 0.656±0.009 | 0.665±0.010 | 0.682±0.009 | 0.506±0.010 | 0.517±0.010 | 0.529±0.011 | 0.411±0.012 | 0.373±0.012 | 0.388±0.011 |
| | MERIT | 0.910±0.005 | 0.659±0.010 | 0.667±0.009 | 0.684±0.010 | 0.508±0.011 | 0.519±0.010 | 0.531±0.010 | 0.413±0.012 | 0.374±0.012 | 0.390±0.012 |
| | ARIEL | 0.911±0.004 | 0.660±0.010 | 0.669±0.010 | 0.686±0.010 | 0.511±0.010 | 0.521±0.011 | 0.533±0.010 | 0.415±0.013 | 0.376±0.013 | 0.392±0.011 |
| | SFA | 0.912±0.004 | 0.658±0.009 | 0.668±0.010 | 0.685±0.009 | 0.509±0.010 | 0.520±0.010 | 0.532±0.010 | 0.414±0.012 | 0.375±0.011 | 0.391±0.012 |
| | Sel-Cl | 0.909±0.005 | 0.661±0.009 | 0.670±0.009 | 0.687±0.009 | 0.512±0.010 | 0.522±0.010 | 0.534±0.010 | 0.416±0.012 | 0.377±0.011 | 0.393±0.011 |
| | Jo-SRC | 0.908±0.004 | 0.659±0.009 | 0.668±0.009 | 0.684±0.010 | 0.510±0.010 | 0.520±0.010 | 0.532±0.011 | 0.413±0.012 | 0.374±0.012 | 0.390±0.012 |
| | GRAND+ | 0.913±0.005 | 0.662±0.010 | 0.671±0.010 | 0.688±0.010 | 0.513±0.010 | 0.523±0.010 | 0.535±0.010 | 0.417±0.012 | 0.378±0.012 | 0.394±0.011 |
| | CGNN | 0.911±0.004 | 0.657±0.009 | 0.667±0.010 | 0.683±0.010 | 0.507±0.010 | 0.518±0.011 | 0.530±0.010 | 0.412±0.012 | 0.373±0.012 | 0.389±0.011 |
| | CRGNN | 0.912±0.004 | 0.659±0.010 | 0.669±0.010 | 0.685±0.010 | 0.509±0.011 | 0.521±0.011 | 0.533±0.011 | 0.414±0.012 | 0.375±0.012 | 0.391±0.012 |
| | HONGAT | 0.910±0.004 | 0.655±0.010 | 0.666±0.009 | 0.682±0.010 | 0.505±0.010 | 0.516±0.010 | 0.528±0.010 | 0.410±0.012 | 0.372±0.012 | 0.387±0.012 |
| | **KCR-GCL** | **0.922±0.005** | **0.692±0.013** | **0.708±0.011** | **0.717±0.011** | **0.545±0.014** | **0.558±0.013** | **0.553±0.013** | **0.442±0.014** | **0.408±0.014** | **0.421±0.013** |

that KCR-GCL still outperforms SOTA GCL methods even when the transductive classifiers are employed.

Table 9: Performance comparison for node classification by inductive linear classifier, transductive two-layer GCN classifier, and transductive classifier used in KCR-GCL. The comparisons are performed on Cora.

| Methods | Noise Type | | | | | | | | | |
|---|---|---|---|---|---|---|---|---|---|---|
| | 0 | 40 | | | 60 | | | 80 | | |
| | - | Asymmetric | Symmetric | Attribute | Asymmetric | Symmetric | Attribute | Asymmetric | Symmetric | Attribute |
| SUGRL (original, inductive classifier) | 0.834±0.005 | 0.564±0.011 | 0.674±0.012 | 0.675±0.009 | 0.468±0.011 | 0.552±0.011 | 0.452±0.012 | 0.280±0.012 | 0.381±0.012 | 0.338±0.014 |
| SUGRL + transductive GCN | 0.833±0.006 | 0.562±0.013 | 0.675±0.015 | 0.673±0.012 | 0.470±0.011 | 0.551±0.011 | 0.454±0.012 | 0.280±0.012 | 0.380±0.012 | 0.340±0.014 |
| SUGRL + linear transductive classifier | 0.836±0.007 | 0.568±0.013 | 0.677±0.010 | 0.674±0.011 | 0.472±0.011 | 0.555±0.011 | 0.457±0.012 | 0.284±0.012 | 0.383±0.012 | 0.341±0.014 |
| MERIT (original, inductive classifier) | 0.831±0.005 | 0.560±0.008 | 0.670±0.008 | 0.671±0.009 | 0.467±0.013 | 0.547±0.013 | 0.450±0.014 | 0.277±0.013 | 0.385±0.013 | 0.335±0.009 |
| MERIT + transductive GCN | 0.831±0.007 | 0.562±0.011 | 0.668±0.013 | 0.672±0.014 | 0.466±0.013 | 0.549±0.015 | 0.451±0.016 | 0.276±0.012 | 0.382±0.014 | 0.337±0.013 |
| MERIT + linear transductive classifier | 0.833±0.003 | 0.562±0.014 | 0.673±0.012 | 0.673±0.011 | 0.466±0.015 | 0.546±0.016 | 0.453±0.017 | 0.280±0.016 | 0.386±0.011 | 0.336±0.014 |
| SFA (original, inductive classifier) | 0.839±0.010 | 0.564±0.011 | 0.677±0.013 | 0.676±0.015 | 0.473±0.014 | 0.549±0.014 | 0.457±0.014 | 0.282±0.016 | 0.389±0.013 | 0.344±0.017 |
| SFA + transductive GCN | 0.837±0.013 | 0.565±0.011 | 0.673±0.017 | 0.673±0.018 | 0.474±0.016 | 0.551±0.015 | 0.453±0.018 | 0.277±0.016 | 0.389±0.015 | 0.343±0.019 |
| SFA + linear transductive classifier | 0.841±0.015 | 0.566±0.013 | 0.678±0.014 | 0.679±0.014 | 0.477±0.015 | 0.552±0.012 | 0.456±0.016 | 0.284±0.017 | 0.391±0.015 | 0.348±0.019 |
| LR-GCL | 0.858±0.006 | 0.589±0.011 | 0.713±0.007 | 0.695±0.011 | 0.492±0.011 | 0.587±0.013 | 0.477±0.012 | 0.306±0.012 | 0.419±0.012 | 0.363±0.011 |
| KCR-GCL | **0.861±0.006** | **0.610±0.011** | **0.731±0.007** | **0.715±0.011** | **0.512±0.011** | **0.610±0.013** | **0.500±0.012** | **0.341±0.012** | **0.444±0.012** | **0.390±0.011** |

## C.3 Statistical Significance Analysis

In this section, we compute the p-values from paired t-tests comparing the performance of KCR-GCL and its ablation model, LR-GCL, against the strongest baseline methods to assess the statistical significance of the observed improvements. As shown in Table 10, the p-values for both KCR-GCL and LR-GCL remain consistently below the threshold of $0.05$ across all datasets and noise conditions, thereby confirming that the performance gains over the top baselines are statistically significant.

Table 10: P-values of the t-tests for LR-GCL and KCR-GCL against the top baseline methods under each noise setting on all the benchmark datasets.

| Datasets | Methods | Noise Type | | | | | | | | |
|---|---|---|---|---|---|---|---|---|---|---|
| | | 40 | | | 60 | | | 80 | | |
| | | Asymmetric | Symmetric | Attribute | Asymmetric | Symmetric | Attribute | Asymmetric | Symmetric | Attribute |
| Cora | LR-GCL | 0.038 | 0.024 | 0.021 | 0.035 | 0.021 | 0.041 | 0.025 | 0.028 | 0.031 |
| | KCR-GCL | 0.027 | 0.022 | 0.018 | 0.018 | 0.038 | 0.035 | 0.031 | 0.023 | 0.030 |
| Citeseer | LR-GCL | 0.043 | 0.035 | 0.022 | 0.021 | 0.030 | 0.041 | 0.027 | 0.024 | 0.022 |
| | KCR-GCL | 0.037 | 0.038 | 0.019 | 0.027 | 0.025 | 0.040 | 0.035 | 0.037 | 0.030 |
| PubMed | LR-GCL | 0.028 | 0.043 | 0.030 | 0.026 | 0.027 | 0.043 | 0.036 | 0.040 | 0.042 |
| | KCR-GCL | 0.025 | 0.030 | 0.026 | 0.023 | 0.024 | 0.041 | 0.033 | 0.035 | 0.037 |
| Coauthor-CS | LR-GCL | 0.041 | 0.032 | 0.036 | 0.043 | 0.044 | 0.040 | 0.027 | 0.037 | 0.042 |
| | KCR-GCL | 0.036 | 0.030 | 0.034 | 0.041 | 0.042 | 0.039 | 0.025 | 0.033 | 0.036 |
| ogbn-arxiv | LR-GCL | 0.040 | 0.032 | 0.036 | 0.043 | 0.044 | 0.040 | 0.027 | 0.037 | 0.042 |
| | KCR-GCL | 0.036 | 0.030 | 0.034 | 0.041 | 0.042 | 0.039 | 0.025 | 0.033 | 0.036 |
| Wiki-CS | LR-GCL | 0.044 | 0.035 | 0.033 | 0.028 | 0.034 | 0.041 | 0.036 | 0.040 | 0.042 |
| | KCR-GCL | 0.040 | 0.036 | 0.031 | 0.026 | 0.029 | 0.039 | 0.034 | 0.038 | 0.040 |
| Amazon-Computers | LR-GCL | 0.033 | 0.043 | 0.034 | 0.031 | 0.034 | 0.041 | 0.036 | 0.040 | 0.042 |
| | KCR-GCL | 0.031 | 0.038 | 0.030 | 0.028 | 0.030 | 0.038 | 0.034 | 0.037 | 0.040 |
| Amazon-Photos | LR-GCL | 0.040 | 0.041 | 0.038 | 0.043 | 0.044 | 0.040 | 0.027 | 0.037 | 0.042 |
| | KCR-GCL | 0.037 | 0.039 | 0.036 | 0.041 | 0.042 | 0.039 | 0.025 | 0.033 | 0.036 |
| Texas | LR-GCL | 0.038 | 0.024 | 0.021 | 0.035 | 0.021 | 0.041 | 0.025 | 0.028 | 0.031 |
| | KCR-GCL | 0.027 | 0.022 | 0.018 | 0.018 | 0.038 | 0.035 | 0.031 | 0.023 | 0.030 |
| Chameleon | LR-GCL | 0.044 | 0.035 | 0.033 | 0.028 | 0.034 | 0.041 | 0.036 | 0.040 | 0.042 |
| | KCR-GCL | 0.040 | 0.036 | 0.031 | 0.026 | 0.029 | 0.039 | 0.034 | 0.038 | 0.040 |

## C.4 Sensitivity Analysis on the Hyperparameters

We perform a sensitivity analysis on the weighting parameter $\tau$ associated with the TNN. This analysis is conducted using KCR-GCL on the Coauthor-CS dataset under the setting of semi-supervised node classification with $60\%$ asymmetric label noise. We vary $\tau$ over the set $\{0.1, 0.2, 0.3, 0.4, 0.5, 0.6, 0.7, 0.8, 0.9\}$, and the corresponding classification accuracies are reported in Table 11. The highest accuracy is achieved at $\tau = 0.5$, though KCR-GCL maintains stable and competitive performance across the entire range. Even in the least favorable case, with $\tau = 0.1$, the accuracy declines by only $0.6\%$, highlighting the robustness of the model to variations in $\tau$.

In addition, we conduct an ablation study on the hyperparameter $M$, which is the maximum power in the KCR self-attention. As discussed in Section 4.1, larger values of $M$ allow the attention matrix $\mathbf{B}$ to incorporate higher-order feature propagation through polynomial kernel expansions. We vary

$M$ from 1 to 9, and the corresponding classification accuracies are also reported in Table 11. The results indicate that KCR-GCL is robust to the choice of $M$, with the best performance observed when $M = 3$.

Table 11: Sensitivity analysis on the weighting parameter $\tau$ for the TNN and the maximum power $M$ in the KCR self-attention. The study is performed using KCR-GCL on the Coauthor-CS dataset for semi-supervised node classification under $60\%$ asymmetric label noise.

| $\tau$ | 0.1 | 0.2 | 0.3 | 0.4 | 0.5 | 0.6 | 0.7 | 0.8 | 0.9 |
|---|---|---|---|---|---|---|---|---|---|
| Accuracy | 0.588 | 0.594 | 0.593 | 0.591 | 0.594 | 0.590 | 0.591 | 0.590 | 0.589 |
| $M$ | 1 | 2 | 3 | 4 | 5 | 6 | 7 | 8 | 9 |
| Accuracy | 0.590 | 0.592 | 0.594 | 0.593 | 0.594 | 0.592 | 0.593 | 0.587 | 0.588 |

We also perform an ablation study to examine the influence of the rank parameter $r_0 = \lceil \gamma \min \{N, d\} \rceil$ in the regularization term employed in the training loss of the KCR-GCL. Table 12 shows the classification performance of LR-GCL under various choices of $r_0$. The results indicate that KCR-GCL consistently maintains near-optimal accuracy across a wide range of rank settings, particularly when $\gamma$ is chosen within the interval 0.1 to 0.3.

Table 12: Ablation study on the value of rank $r_0$ in the optimization problem (1) on Cora with different levels of asymmetric and symmetric label noise. The accuracy with the optimal rank is shown in the last row. The accuracy difference against the optimal rank is shown for other ranks. 'Asy' and 'Sy' in this table denote the asymmetric label noise and symmetric label noise.

| | Noise Type | | | | | | |
|---|---|---|---|---|---|---|---|
| $\gamma$ | 0 | 40 | | 60 | | 80 | |
| | - | Asy | Sy | Asy | Sy | Asy | Sy |
| 0.1 | -0.002 | -0.001 | -0.002 | -0.002 | -0.001 | -0.001 | -0.000 |
| 0.2 | -0.000 | -0.000 | -0.000 | -0.000 | -0.000 | -0.000 | -0.000 |
| 0.3 | -0.000 | -0.000 | -0.001 | -0.002 | -0.001 | -0.000 | -0.001 |
| 0.4 | -0.001 | -0.003 | -0.002 | -0.001 | -0.002 | -0.002 | -0.002 |
| 0.5 | -0.001 | -0.002 | -0.003 | -0.003 | -0.003 | -0.001 | -0.002 |
| 0.6 | -0.003 | -0.002 | -0.002 | -0.003 | -0.002 | -0.002 | -0.003 |
| 0.7 | -0.003 | -0.004 | -0.003 | -0.004 | -0.004 | -0.004 | -0.005 |
| 0.8 | -0.002 | -0.005 | -0.006 | -0.006 | -0.006 | -0.007 | -0.007 |
| 0.9 | -0.004 | -0.004 | -0.005 | -0.007 | -0.008 | -0.008 | -0.006 |
| 1.0 | -0.004 | -0.004 | -0.007 | -0.007 | -0.008 | -0.010 | -0.008 |
| optimal | 0.858 | 0.589 | 0.713 | 0.492 | 0.587 | 0.306 | 0.419 |

Table 13: Comparisons in the kernel complexity defined in Theorem 4.1 of the main paper. The evaluation is performed on the semi-supervised node classification task with $40\%$ of symmetric label noise.

| Datasets | | MERIT | SFA | Jo-SRC | GCN | GFSA | HONGAT | LR-GCL | KCR-GCL |
|---|---|---|---|---|---|---|---|---|---|
| Cora | KC | 0.37 | 0.42 | 0.48 | 0.44 | 0.35 | 0.40 | 0.20 | 0.14 |
| | $r_0$ | 1420 | 1478 | 1665 | 1511 | 1262 | 1450 | 440 | 395 |
| Citeseer | KC | 0.47 | 0.45 | 0.55 | 0.64 | 0.47 | 0.50 | 0.24 | 0.18 |
| | $r_0$ | 1214 | 1180 | 1405 | 1590 | 1224 | 1285 | 405 | 369 |
| PubMed | KC | 0.54 | 0.50 | 0.62 | 0.71 | 0.52 | 0.66 | 0.30 | 0.25 |
| | $r_0$ | 1644 | 1562 | 1785 | 1993 | 1588 | 1874 | 1197 | 1090 |
| Wiki-CS | KC | 0.42 | 0.44 | 0.40 | 0.49 | 0.43 | 0.45 | 0.19 | 0.14 |
| | $r_0$ | 1805 | 1993 | 1746 | 2130 | 1842 | 2048 | 970 | 904 |
| Amazon-Computers | KC | 0.39 | 0.37 | 0.40 | 0.45 | 0.35 | 0.37 | 0.12 | 0.10 |
| | $r_0$ | 1450 | 1428 | 1489 | 1632 | 1370 | 1415 | 874 | 820 |
| Amazon-Photos | KC | 0.38 | 0.38 | 0.43 | 0.47 | 0.39 | 0.41 | 0.14 | 0.12 |
| | $r_0$ | 1872 | 1884 | 1990 | 2145 | 1895 | 1921 | 750 | 722 |
| Coauthor-CS | KC | 0.29 | 0.28 | 0.32 | 0.34 | 0.31 | 0.32 | 0.12 | 0.10 |
| | $r_0$ | 1774 | 1725 | 1896 | 1903 | 1872 | 1890 | 1120 | 1039 |
| ogbn-arxiv | KC | 0.12 | 0.13 | 0.12 | 0.14 | 0.12 | 0.13 | 0.05 | 0.04 |
| | $r_0$ | 1860 | 1936 | 1852 | 1996 | 1845 | 1920 | 1354 | 1328 |

## C.5 ADDITIONAL STUDY IN THE KERNEL COMPLEXITY

We further assess the kernel complexity (KC) of the gram matrix computed from node representations generated by KCR-GCL, its ablation model LR-GCL, and several competing baseline methods

across more benchmark datasets. This evaluation is conducted under symmetric label noise with a corruption rate of $40\%$. As shown in Table 13, the node representations learned by KCR-GCL exhibit consistently lower kernel complexity, suggesting that transductive classifiers trained on such representations are likely to achieve smaller generalization errors on previously unseen nodes.

### C.6 EIGEN-PROJECTION AND CONCENTRATION ENTROPY ANALYSIS ON ADDITIONAL DATASETS

Figure 2 illustrates the eigen-projection visualizations and corresponding signal concentration ratios for Coauthor-CS, Amazon-Computers, Amazon-Photos, and ogbn-arxiv. We also investigate the presence of the Low Frequency Property (LFP) in heterophilic graph benchmarks, Texas and Chameleon (Pei et al., 2020), through eigen-projection plots and signal concentration ratio analysis, as illustrated in Figure 3. The findings reveal that the LFP persists in heterophilic settings, similar to homophilic graphs. The analysis is performed under asymmetric label noise with a corruption rate of $60\%$. When setting the rank parameter to $0.2 \min\{N, d\}$, the corresponding concentration entropy scores are observed to be $0.762$ for Chameleon and $0.725$ for Texas.

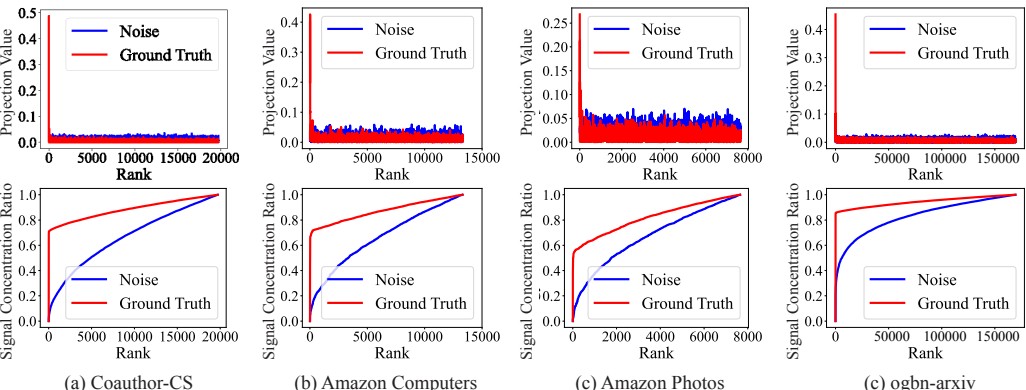

(a) Coauthor-CS     (b) Amazon Computers     (c) Amazon Photos     (c) ogbn-arxiv

Figure 2: Eigen-projection (first row) and energy concentration (second row) on Coauthor-CS, Amazon-Computers, Amazon-Photos, and ogbn-arxiv. By the rank of $0.2 \min\{N, d\}$, the concentration entropy on Coauthor-CS, Amazon-Computers, Amazon-Photos, and ogbn-arxiv are $0.779$, $0.809$, $0.752$, and $0.787$.

**Study of the Low Frequency Property (LFP) for Attribute Noise.** We study how the information from the ground-truth labels is distributed across different eigenvectors of the feature gram matrix $\mathbf{K_F}$ when the feature $\mathbf{F}$ is learned from the graph with attribute noise. It is noted that the observed label $\mathbf{Y} \in \mathbb{R}^{N \times C}$ is the clean ground-truth label without any noise in this setting. Following Figure 1 in Section 4.1, we compute the eigen-projection score of the label $\mathbf{Y}$ on the eigenvectors of the gram matrix $\mathbf{K_F}$ and the corresponding signal concentration ratios. Figure 4 illustrates that the ground-truth label signals are primarily concentrated on the leading eigenvectors of $\mathbf{K_F}$, even when the feature $\mathbf{F}$ is learned from the graph with attribute noise. The above observation motivates learning low-rank features for node classification with attribute noise.

### C.7 TRAINING TIME COMPARISON

In this section, we report a comparative analysis of the training time for KCR-GCL and other baseline methods across all benchmark datasets. The total training time for LR-GCL encompasses three components: the time required for robust graph contrastive learning, the computation time for the singular value decomposition (SVD) of the kernel matrix, and the training time of the transductive classifier.

For the baseline GCL methods, the reported training time includes both the encoder training phase and the downstream classifier training. All experiments are conducted using a single 80 GB NVIDIA A100 GPU. The detailed results are provided in Table 14. As shown, the overall training time of KCR-GCL is comparable to that of state-of-the-art GCL methods such as SFA and MERIT.

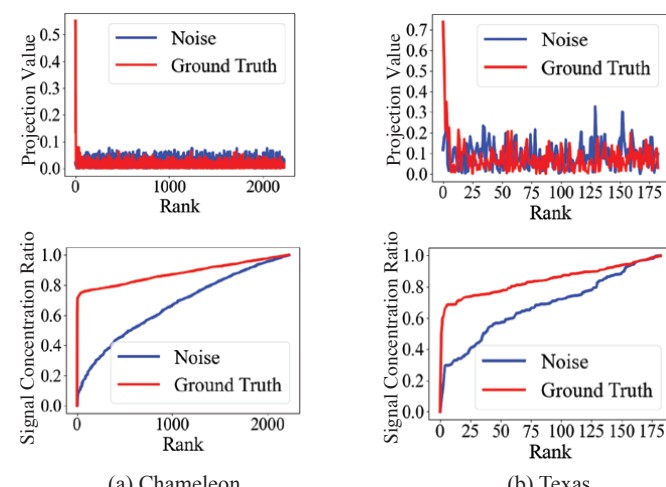

(a) Chameleon      (b) Texas

Figure 3: Eigen-projection (first row) and signal concentration ratio (second row) on Chameleon and Texas. The study in this figure is performed for asymmetric label noise with a noise level of 60%. By the rank of $0.2 \min \{N, d\}$, the concentration entropy on Chameleon and Texas are $0.762$ and $0.725$.

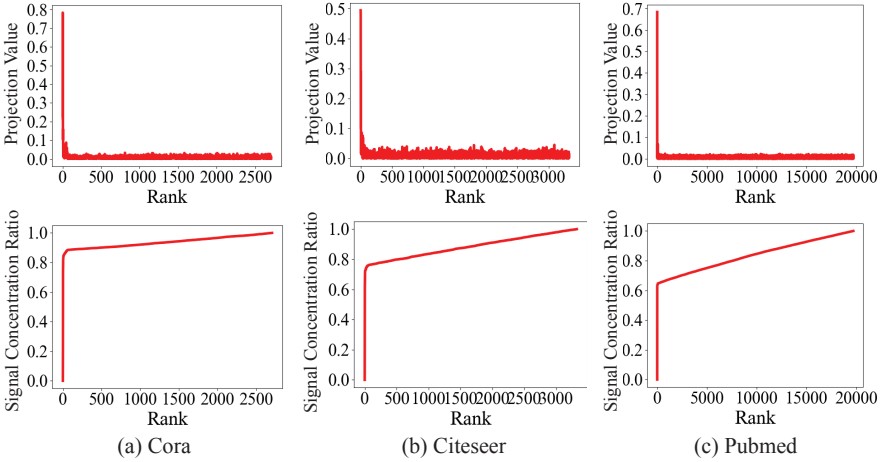

(a) Cora      (b) Citeseer      (c) Pubmed

Figure 4: Eigen-projection (first row) and signal concentration ratio (second row) on Cora, Citeseer, and Pubmed, as the illustration of the Low Frequency Property (LFP). The study in this figure is performed on graphs consisting of attribute noise with a noise level of 60%. By the rank $r = 0.2 \min \{N, d\}$, the signal concentration ratio of $\tilde{\mathbf{Y}}$ for Cora, Citeseer, and Pubmed are $0.815$, $0.785$, and $0.689$ respectively.

## D  ALGORITHMS

Algorithm 1 presents the training procedure for the Prototypical Graph Contrastive Learning (GCL) encoder. At each iteration, the model computes node representations via the current GCN encoder, clusters them into $K$ semantic prototypes, and generates augmented graph views. The model parameters $\theta$ are then updated by minimizing the sum of the node-level and prototype-level contrastive losses. This iterative process encourages the encoder to learn robust and semantically meaningful node embeddings through prototype-guided alignment.

Algorithm 2 details the full training procedure for KCR-GCL. Starting from the GCL encoder pre-trained by GCL, the algorithm iteratively updates the encoder parameters $\theta$, the classifier weights $\mathbf{W}$, and the KCR self-attention weights $\alpha$. At each training step, the encoder is first updated by

Table 14: Training time (seconds) comparisons for node classification.

| Methods | Cora | Citeseer | PubMed | Coauthor CS | Wiki-CS | Amazon Computer | Amazon Photo | ogbn-arxiv |
|---------|------|----------|--------|-------------|---------|-----------------|--------------|------------|
| GCN | 11.5 | 13.7 | 38.6 | 43.2 | 22.3 | 30.2 | 19.0 | 215.1 |
| $S^2$GC | 20.7 | 22.5 | 47.2 | 57.2 | 27.6 | 38.5 | 22.2 | 243.7 |
| GCE | 32.6 | 36.9 | 67.3 | 80.8 | 37.6 | 50.1 | 32.2 | 346.1 |
| NRGNN | 72.4 | 80.5 | 142.7 | 189.4 | 74.3 | 97.2 | 62.4 | 650.2 |
| RTGNN | 143.3 | 169.5 | 299.5 | 353.5 | 153.7 | 201.5 | 124.2 | 1322.2 |
| SUGRL | 100.3 | 122.1 | 207.4 | 227.1 | 107.7 | 142.8 | 87.7 | 946.8 |
| MERIT | 167.2 | 179.2 | 336.7 | 375.3 | 172.3 | 226.5 | 140.6 | 1495.1 |
| ARIEL | 156.9 | 164.3 | 284.3 | 332.6 | 145.1 | 190.4 | 118.3 | 1261.4 |
| SFA | 237.5 | 269.4 | 457.1 | 492.3 | 233.5 | 304.5 | 187.2 | 2013.1 |
| Sel-Cl | 177.3 | 189.9 | 313.5 | 352.5 | 161.7 | 211.1 | 130.9 | 1401.1 |
| Jo-SRC | 148.2 | 157.1 | 281.0 | 306.1 | 144.5 | 188.0 | 118.5 | 1256.0 |
| GRAND+ | 57.4 | 68.4 | 101.7 | 124.2 | 54.8 | 73.8 | 44.5 | 479.2 |
| LR-GCL | 159.9 | 174.5 | 350.7 | 380.9 | 180.3 | 235.7 | 145.5 | 1552.7 |
| KCR-GCL | 166.2 | 185.4 | 372.7 | 399.5 | 195.4 | 253.6 | 159.2 | 1674.8 |

optimizing the contrastive loss. Then, based on the updated encoder, node representations are re-computed and used to optimize the KCR-GCL objective $L(\mathbf{W}, \boldsymbol{\theta}, \boldsymbol{\alpha})$, which includes both classification loss and low-rank regularization. This two-stage update ensures that the final representations achieve both semantic alignment and low kernel complexity.

---

**Algorithm 1** Training of the Prototypical Graph Contrastive Learning (PGCL)

---

1: **Input:** Attribute matrix $\mathbf{X}$, adjacency matrix $\mathbf{A}$, training epochs $t_{\max}$, learning rate $\eta$
2: **Output:** The GCL encoder $g_{\boldsymbol{\theta}}$
3: Randomly initialize model parameters $\boldsymbol{\theta}^{(0)}$
4: **for** $t = 1$ to $t_{\max}$ **do**
5:     Compute node representations $\mathbf{H} = g_{\boldsymbol{\theta}^{(t-1)}}(\mathbf{X}, \mathbf{A})$
6:     Cluster $\{\mathbf{H}_i\}_{i=1}^N$ into $K$ clusters $\{S_k\}_{k=1}^K$ using $K$-means
7:     **for** $k = 1$ to $K$ **do**
8:         Compute prototype $\mathbf{c}_k = \frac{1}{|S_k|} \sum_{\mathbf{H}_i \in S_k} \mathbf{H}_i$
9:     **end for**
10:    Generate augmented views $G^1 = (\mathbf{X}^1, \mathbf{A}^1)$ and $G^2 = (\mathbf{X}^2, \mathbf{A}^2)$
11:    Compute $\mathbf{H}^1 = g_{\boldsymbol{\theta}^{(t-1)}}(\mathbf{X}^1, \mathbf{A}^1)$, $\mathbf{H}^2 = g_{\boldsymbol{\theta}^{(t-1)}}(\mathbf{X}^2, \mathbf{A}^2)$
12:    Perform gradient descent on $\boldsymbol{\theta}$ by $\boldsymbol{\theta}^{(t)} \leftarrow \boldsymbol{\theta}^{(t-1)} - \eta \nabla_{\boldsymbol{\theta}} \mathcal{L}_{\text{GCL}}(\boldsymbol{\theta}^{(t-1)})$
13: **end for**
14: **Return** the GCL encoder $g_{\boldsymbol{\theta}^{(t_{\max})}}$

---

---

**Algorithm 2** Training Algorithm for the KCR-GCL Encoder

---

1: **Input:** Attribute matrix $\mathbf{X}$, adjacency matrix $\mathbf{A}$, training epochs $t_{\max}$, parameters of the pre-trained GCL encoder $\boldsymbol{\theta}^{(0)}$, learning rate $\eta$
2: **Output:** The GCL encoder $g_{\boldsymbol{\theta}}$, the weights of the classifier $\mathbf{W}$, and the weights of the KCR self-attention $\boldsymbol{\alpha}$
3: Randomly initialize $\mathbf{W}^{(0)}$ and $\boldsymbol{\alpha}^{(0)}$
4: **for** $t = 1$ to $t_{\max}$ **do**
5:     Compute node representations $\mathbf{H} = g_{\boldsymbol{\theta}^{(t-1)}}(\mathbf{X}, \mathbf{A})$
6:     Cluster $\{\mathbf{H}_i\}_{i=1}^N$ into $K$ clusters $\{S_k\}_{k=1}^K$ using $K$-means
7:     **for** $k = 1$ to $K$ **do**
8:         Compute prototype $\mathbf{c}_k = \frac{1}{|S_k|} \sum_{\mathbf{H}_i \in S_k} \mathbf{H}_i$
9:     **end for**
10:    Generate augmented views $G^1 = (\mathbf{X}^1, \mathbf{A}^1)$ and $G^2 = (\mathbf{X}^2, \mathbf{A}^2)$
11:    Compute $\mathbf{H}^1 = g_{\boldsymbol{\theta}^{(t-1)}}(\mathbf{X}^1, \mathbf{A}^1)$, $\mathbf{H}^2 = g_{\boldsymbol{\theta}^{(t-1)}}(\mathbf{X}^2, \mathbf{A}^2)$
12:    Perform gradient descent on $\boldsymbol{\theta}$ and $\boldsymbol{\alpha}$ by $\boldsymbol{\theta}^{(t)} \leftarrow \boldsymbol{\theta}^{(t-1)} - \eta \nabla_{\boldsymbol{\theta}} \mathcal{L}_{\text{KCR-GCL}}(\boldsymbol{\theta}^{(t-1)}, \boldsymbol{\alpha}^{(t-1)})$ and
      $\boldsymbol{\alpha}^{(t)} \leftarrow \boldsymbol{\alpha}^{(t-1)} - \eta \nabla_{\boldsymbol{\alpha}} \mathcal{L}_{\text{KCR-GCL}}(\boldsymbol{\theta}^{(t-1)}, \boldsymbol{\alpha}^{(t-1)})$.
13: **end for**
14: Compute node representations $\mathbf{F} = \mathbf{B}\mathbf{H} = \mathbf{B}g_{\boldsymbol{\theta}^{(t)}}(\mathbf{X}, \mathbf{A})$.
15: **Return** the KCR-GCL encoder, and the node representations $\mathbf{F}$.

---

