# OpenReview forum: "Kernel Complexity Reduced Graph Contrastive Learning for Noisy Node Classification"
_ICLR.cc/2026/Conference — ICLR 2026 Conference Withdrawn Submission_

### Official Review · Reviewer_M5Tm · 2025-10-30

**Soundness:** 3
**Presentation:** 3
**Contribution:** 3
**Rating:** 6
**Confidence:** 3

**Summary:**

The paper introduces the low-rank regularization to node representation learning in the graph constrastive learning setting, with theoretical analysis of the generalization bound. The node classification experiments on several homophilious graphs and two heterophilic graphs are performed to demonstrate the merits of the proposed method.

**Strengths:**

* The redundancy of correlations among node features in graph self-supervised learning is explored and mitigated through a Gram matrix–based low-rank regularizer.
* The problem is well motivated by analyzing eigen-projection and signal concentration ratio changes under label noise.
* Experimental results demonstrate the effectiveness and robustness of KCR-GCL in learning node representations against random attribute and label perturbations.

**Weaknesses:**

* Only two heterophilic graphs are used for evaluation, and one of them, Chameleon, is problematic due to a large number of duplicate nodes. It is recommended to include more heterophilic graphs for a more comprehensive evaluation.

*  Robustness experiments against structural perturbations (e.g., Metattack or Nettack) are missing.

* It would also be beneficial to evaluate the proposed model on graph classification tasks, which are commonly used benchmarks for graph contrastive learning.

* Although the low-rank regularization is shown to be effective and empirical studies confirm the low-rank property among node features, a deeper explanation of how this regularization contributes to model performance would strengthen the work.

**Questions:**

1) In KCR-GCL, the prototypical contrastive learning (PCL) module is incorporated to align node representations with their corresponding cluster centers. This component may also contribute to the robustness of node representation learning against noise. It would be interesting to analyze the relative importance of PCL and KCR — which component plays a more significant role in achieving robustness?

2) Furthermore, are there any conceptual or empirical connections between the Gram matrix–based low-rank regularization and node clustering? Could the clustering effect of GCL' node embeddings be further enhanced through this low-rank constraint?

3) It also appears that KCR may not be inherently specific to graph contrastive learning (GCL). Would the proposed design remain effective under other learning paradigms, such as supervised learning?

4)Finally, the paper mentions that the “graph” is low-rank (e.g., in the abstract). It would be helpful to clarify whether “graph” here refers to the correlation matrix K that defines node relationships, or to the original input graph structure.

---

### Official Review · Reviewer_DzfD · 2025-10-31

**Soundness:** 2
**Presentation:** 2
**Contribution:** 2
**Rating:** 4
**Confidence:** 3

**Summary:**

This paper introduces KCR-GCL, which learns robust node representations and tackles noisy node classification. Experiments on various benchmarks highlight the effectiveness and robustness in learning node representations under noisy conditions.

**Strengths:**

1. The paper provides a theoretical analysis that establishes a generalization guarantee for the linear transductive classifier trained on the low-rank node representations.

2. The experimental setup is exhaustive, involving multiple baselines, datasets, and comprehensive analyses.

**Weaknesses:**

1. Inconsistent KCR-GCL scores between Table 5 and Table 7.

2. Complex Hyperparameter Tuning. Multiple hyperparameters (the weighting parameter, the maximum power,and  the rank parameter) require careful tuning, increasing the tuning burden.

**Questions:**

1. Do the selected baselines cover loss correction, sample selection, label denoising, structural regularization, and auxiliary self-supervised methods?

2. Are there any real-world noisy datasets, instead of those created by injecting synthetic noise into clean datasets?

---

### Official Review · Reviewer_EX6T · 2025-11-01

**Soundness:** 2
**Presentation:** 3
**Contribution:** 2
**Rating:** 2
**Confidence:** 3

**Summary:**

This paper studied noise in graph data, included in both node attributes and labels. The theoretical support is to reduce the kernel complexity of the emebeding gram matrix. Therefore, the authors proposed two ways, 1) add low-rank loss, and 2) propagate output embeddings over their similarity matrix. Extensive experiments show the good performances of the proposed model.

**Strengths:**

1. Studying how to alleviate noise in graph data is a good motivation.
2. This paper has theoretical support.
3. The authors did extensive experiments.

**Weaknesses:**

1. The core property, i.e., Low Frequency Property, actually is not new, also the authors acknowledged this point. Two techniques that lowing ranks essentially enhance low frequency components.
2. The whole pipeline faces a huge computing complexity. 1) B has shape $N\times N$, 2) $F=BH$ where B is a dense matrix with O(N^2), 3) EVD on $K$ to get eigenvalues with O(N^3). All these three parts are done once at each epoch.
3. I cannot get why the authors adopt this technique on contrastive learning based models, rather than traditional semi-supervised GNNs. The proposed techniques are not specific to contrastive learning actually.
4. In Table 1, although the proposed KCR-GCL always performed the best, its performances also decreased with the increase of noise ratio, nearly at the same decreasing rate as baselines. This cannot prove the robustness of KCR-GCL. Robustness means you can perform well at clean, as well as not decrease too much like others at noise.

**Questions:**

According to Theorem 4.1, the loss is upper bounded by L1, L2 and KC. Actually, usage of B can decrease KC, but how can it guarantee a lower L1 and L2? I notice that in Table 3, L1 and L2 were lower for KCR-GCL, but can we have some theoretical intuitions? There may be bias in experimental results.

---

### Official Review · Reviewer_CsxG · 2025-11-01

**Soundness:** 3
**Presentation:** 3
**Contribution:** 2
**Rating:** 4
**Confidence:** 4

**Summary:**

This paper, inspired by the spectral insight that clean label information tends to concentrate in low-frequency components, proposes the KCR-GCL method and provides corresponding theoretical support, demonstrating that it can learn more robust and generalizable representations for graph node classification tasks.

**Strengths:**

- The paper is well-written and easy to follow.
- The paper provides a generalization bound for learning with noisy data from the perspective of kernel complexity, offering solid theoretical grounding.
- The designs of TNN and KCR Self-Attention are reasonable, with clear motivation and implementation logic.

**Weaknesses:**

- The assumptions are not sufficiently general. Although informative features often lie in low-frequency components, in sparse or heterophilic graphs, high-frequency features can be critical discriminative signals. Thus, the low-rank–encouraging TNN regularization may have limited generality.
- The computational cost is high. Calculating the matrix B is inefficient and expensive, making the method difficult to scale to large-scale graphs such as OGB datasets.
- Although the authors provide some explanations, the theoretical proof is still based on MSE loss and assumes a large number of labeled nodes, which is inconsistent with the cross-entropy loss used in actual training.
- In my understanding, the hyperparameters $\tau$ and $r_{0}$ are not easy to determine and may cause training instability.

**Questions:**

- Have you considered conducting experiments on larger-scale datasets?
- Have you considered using real noisy graph datasets (e.g., NoisyGL) instead of synthetic noise generation?
- Please address the concerns raised in the Weaknesses section.

---

### Note · Authors · 2026-01-09

I have read and agree with the venue's withdrawal policy on behalf of myself and my co-authors.